# Improving the Straight-Through Estimator with Zeroth-Order Information

**Ningfeng Yang**
University of British Columbia
nxyang@ece.ubc.ca

**Tor M. Aamodt**
University of British Columbia
aamodt@ece.ubc.ca

## Abstract

We study the problem of training neural networks with quantized parameters. Learning low-precision quantized parameters by enabling computation of gradients via the Straight-Through Estimator (STE) can be challenging. While the STE enables back-propagation, which is a first-order method, recent works have explored the use of zeroth-order (ZO) gradient descent for fine-tuning. We note that the STE provides high-quality biased gradients, and ZO gradients are unbiased but can be expensive. We thus propose First-Order-Guided Zeroth-Order Gradient Descent (FOGZO) that reduces STE bias while reducing computations relative to ZO methods. Empirically, we show FOGZO improves the tradeoff between quality and training time in Quantization-Aware Pre-Training. Specifically, versus STE at the same number of iterations, we show a 1-8% accuracy improvement for DeiT Tiny/Small, 1-2% accuracy improvement on ResNet 18/50, and 1-22 perplexity point improvement for LLaMA models with up to 0.3 billion parameters. For the same loss, FOGZO yields a $796\times$ reduction in computation versus n-SPSA for a 2-layer MLP on MNIST. Code is available at `https://github.com/1733116199/fogzo`.

## 1 Introduction

Quantization is an effective method to reduce the compute and memory complexity of Deep Neural Networks (DNNs), enabling DNNs to be deployed to resource-constrained platforms. However, naively applying quantization often leads to detrimental accuracy degradation. While Post-Training Quantization (PTQ) (23; 43; 29; 24; 20) methods can mitigate such accuracy degradation and produce high-quality 4+-bit models with little computation and memory overhead, Quantization-Aware Training (QAT) (35; 13; 28; 21; 12) methods can produce even more accurate models at lower precisions by integrating quantization into the training loop. Lower-precision models can be important in edge applications where large hardware deployments are expensive to change. As such, this work focuses on improving QAT.

QAT methods can be further taxonomized into Quantization-Aware Pre-Training (QAPT) (35; 21) and Quantization-Aware Fine-Tuning (QAFT) (30; 12; 54). QAFT methods initialize a low-precision model from a full-precision pre-trained checkpoint and subsequently fine-tune the model for a short duration (12) on a small downstream dataset. On the other hand, QAPT methods randomly initialize a low-precision model and pre-train it for long durations on large-scale pre-training datasets (35). Compared to first pre-training in full precision and subsequently applying PTQ/QAFT, QAPT can have higher pre-training speed and lower pre-training cost, as when weights and activations are quantized, the forward pass may be computed in low precision (21; 51). This work aims to improve the accuracy/prediction quality of QAPT.

A core challenge of QAT is the non-differentiability of the objective function. The rounding function used to discretize weights/activations has zero gradient almost everywhere and is non-differentiable at rounding boundaries. A gradient that is zero almost everywhere suggests parameters remain

39th Conference on Neural Information Processing Systems (NeurIPS 2025).

unchanged during gradient descent. To work around this, the Straight-Through Estimator (19; 3) (STE) is a well-known method to enable BackProp through non-differentiable operations such as rounding, as if they were the identity function or other smooth functions (52). At high precision, the STE performs surprisingly well (3) despite having little theoretical justification (52; 30). However, at low precision, the STE does not work well (33; 25). On the other hand, zeroth-order (ZO) methods, which estimate gradients using finite differences, while more theoretically sound (3) are computationally intensive and may yield inferior-quality results for a fixed training budget.

We discuss the strengths and weaknesses of the STE and ZO methods and propose FOGZO, which uses the efficiency of the STE to accelerate n-SPSA (45; 46; 31), a ZO method. Our heuristic derivation suggests that FOGZO can suppress the STE gradient when it points in an incorrect direction. Experiments across a wide variety of benchmarks suggest that FOGZO can outperform the STE, while requiring less computation than n-SPSA.

## 2 Background and Motivation

Consider a neural network with continuous parameters $\theta$ and objective function $L(\theta)$. While the loss depends on them as well, for notational convenience, we omit the network input, output, and dataset. To quantize/discretize $\theta$, we augment the objective function to indicate a quantized loss function $\hat{L}(\theta)$. We focus on parameter (as opposed to activation) quantization; thus, $\hat{L}$ can be expressed as:

$$\hat{L}(\theta) = L(\hat{\theta}) \text{ where } \hat{\theta} \text{ is a vector of quantized parameters}$$

$$\hat{\theta} = \begin{cases} f(\text{sign}(f^{-1}(\theta)) & \text{, for binarization} \\ f(\text{clip}(\text{round}(f^{-1}(\theta)))) & \text{, for multi-bit quantization} \end{cases} \tag{1}$$

where $f(\psi) = \alpha\psi$ is an affine dequantization function that converts discretized weights $\psi$ back to their original domain[1] with a quantization scale $\alpha$, and $f^{-1}$ is the inverse of $f$. The functions sign, clip, and round handle the discretization. During QAT, $\hat{L}$ is optimized[2] by varying $\theta$. However, when BackProp is naively applied, the gradient $\frac{\partial \hat{L}}{\partial \theta}$ is always zero because of the existence of non-differentiable operations like sign and round, which are flat almost everywhere and non-differentiable at rounding boundaries.

In this section, we discuss existing methods that address the non-differentiability of $\hat{L}$, including first-order methods like the Straight-Through Estimator and zeroth-order methods like n-SPSA. We analyze the strengths and weaknesses of both methods, and finally discuss biased gradient estimators which can potentially address the weaknesses of zeroth-order methods.

### 2.1 The STE: Empirically Good but Theoretically Questionable

The STE circumvents the non-differentiability of the round and sign functions by replacing their Jacobian with the Jacobian of a smooth approximation. We denote these approximations as smooth_round and smooth_sign, respectively. For example, to perform BackProp on Equation 1, we use the approximation shown in Equation 2:

$$\frac{\partial \hat{L}}{\partial \theta} = \frac{\partial L}{\partial \hat{\theta}} \frac{\partial \hat{\theta}}{\partial \theta}, \text{ where } \frac{\partial \hat{\theta}}{\partial \theta} \approx \begin{cases} \frac{\partial f}{\partial \text{sign}} \cdot \frac{\partial \text{smooth\_sign}}{\partial f^{-1}} \cdot \frac{\partial f^{-1}}{\partial \theta} & \text{, for binarization} \\ \frac{\partial f}{\partial \text{clip}} \cdot \frac{\partial \text{clip}}{\partial \text{round}} \cdot \frac{\partial \text{smooth\_round}}{\partial f^{-1}} \cdot \frac{\partial f^{-1}}{\partial \theta} & \text{, for multi-bit quantization} \end{cases} \tag{2}$$

The most famous example of smooth_round is the identity function (3), but there are alternatives like Soft Rounding (2), HTGE (36), and Confidence-Guided Masking (25). For smooth_sign, popular choices include Hardtanh (38), tanh (38), or ApproxSign (26). The smoothness of these approximations ensures the gradient is not zero everywhere, and therefore training can make progress. The STE is easy to implement, as it only requires changing the backward pass of round/sign operators. The STE has the same computational complexity as regular training, as it only requires $O(1)$ operations for the backward pass of every weight/activation. Despite the simplicity of the STE, it works surprisingly well across a variety of benchmarks (3; 13; 4; 28) and remains the most popular method for QAT.

---

[1]Note that other formulations of $f$ are possible, such as $f(\psi) = \alpha(\psi - z)$ where $z$ is a zero point, or $f(\psi) = M\psi$ where $M$ is an outlier smoothing matrix like the Hadamard matrix.

[2]The value of $\theta$ that minimizes $\hat{L}$ may differ from that which minimizes $L$.

Despite its empirical success and popularity, the theoretical optimality of the STE has remained a topic of debate. While some believe the STE is suboptimal and propose workarounds like Coordinate Descent (30), Stochastic Rounding (17), and Pseudo-Quantization Noise (44), others provide theoretical justification for using the STE on shallow networks and suggest potential for extending their theoretical insights to deeper networks (52). While some observe that specific choices of smooth approximations work better for specific benchmarks (50; 38; 26), others suggest all smooth approximations are equally optimal, and that any observed differences are simply due to implicit weight initialization strategies and learning rate schedules (42). While the STE works well for high-precision networks, it has been shown (33; 25) to introduce parameter oscillations at rounding boundaries, which slows down convergence and reduces the efficiency of training.

Based on the above observations, we conjecture that the STE is a "sufficiently good" gradient estimator that generalizes to a wide variety of techniques, networks, and datasets. However, the STE occasionally produces gradients with "wrong" directions (see Section A). This can contribute to undesirable behaviour (such as the above-noted parameter oscillations (33; 25)). As elaborated below, if we can correct the occasional "flaws" of the STE, we can achieve better-than-STE training accuracy without sacrificing ease of implementation and compute efficiency.

## 2.2 Theoretically Sound but Expensive Alternatives to the STE: Zeroth-Order Methods

Why has the STE stood the test of time and remained the most popular gradient estimator for 13 years, despite being theoretically questionable? We believe this is because the theoretically sound alternative, known as zeroth-order (ZO) methods, is impractical for deep learning. ZO methods operate by perturbing weights and observing changes in loss. For example, Ghadimi et al. (15) and Gasnikov et al. (14) showcase that, given an arbitrary non-differentiable function $\hat{L}$, it is possible to construct a randomly smoothed approximation $\hat{L}_{\text{smooth}}$ using Equation 3, where $\epsilon$ is a positive hyperparameter that represents the smoothness, and $p(u)$ is a perturbation probability distribution[3] that is typically symmetric, zero-mean, and unit-variance/isotropic. Once we have the smooth approximation $\hat{L}_{\text{smooth}}$, its gradient can then be numerically measured using Equation 4, also known as n-SPSA (31; 47), Randomized Gradient Estimator (54; 34), or the zeroth-order gradient estimator (27; 1).

$$\hat{L}_{\text{smooth}}(\theta) = E_{u \sim p(u)}[\hat{L}(\theta + \epsilon u)] \tag{3}$$

$$\nabla \hat{L}_{\text{smooth}}(\theta) \approx \frac{1}{n} \sum_{i=1}^{n} \frac{\hat{L}(\theta + \epsilon u_i) - \hat{L}(\theta - \epsilon u_i)}{2\epsilon} u_i \text{ where } u_i \sim p(u) \tag{4}$$

In n-SPSA, the variance of the measured gradient correlates with the sample size $n$. In an ideal world where we can afford infinite computation, or as $n \to \infty$, we can very accurately measure $\nabla \hat{L}_{\text{smooth}}(\theta)$ with zero variance. In such a case, n-SPSA can outperform the STE. Intuitively, if training with $\nabla \hat{L}_{\text{smooth}}$ can reduce $\hat{L}_{\text{smooth}}$, it will also reduce $\hat{L}$ by extension, since $\hat{L}_{\text{smooth}}(\theta) \approx \hat{L}(\theta)$ for all $\theta$. However, in a practical system, using a large $n$ can be prohibitively expensive, as it requires evaluating the loss $2n$ times. Therefore, reducing $n$ when computation is a limited resource introduces large variance into the measured gradient, which hinders fast convergence. As such, ZO methods for deep learning (54; 31; 6) either underperform compared to first-order methods or only work for specific training scenarios (e.g., fine-tuning).

This presents a dilemma. While n-SPSA has stronger theoretical guarantees than the STE, without sufficient computation, n-SPSA converges more slowly due to its large gradient variance. Is there a way we can reduce the computation while still mitigating gradient variance?

## 2.3 Biased Gradient Estimators: A Potential to Reduce Computation

A method to reduce computation or increase efficiency without increasing gradient variance is to intentionally introduce bias into the gradient sampling process (5). Unlike n-SPSA, which is unbiased and relies solely on a stochastic oracle, biased gradient estimators rely on a source of bias (9). Bias can be harmful but also potentially helpful if the to-be-estimated gradient is likely pointing in a similar direction and has a similar magnitude as that bias. A good source of bias typically has lower variance than unbiased estimators and is more efficient to compute. With a carefully selected bias

---

[3]In some literature, $p(u)$ is known as a *stochastic oracle*. A typical choice of $p(u)$ is the standard normal.

source, biased gradient estimators can potentially match the performance of unbiased ones while using less computation (11).

Given this insight, we ask the research question: *Is it possible to combine the STE, ZO methods, and biased gradient estimators to create a method that outperforms the STE, retains some theoretical guarantees of ZO methods, while being computationally cheap like biased gradient estimators?* In particular, is it possible for the proposed method to outperform the STE across a variety of benchmarks while still being practical for deep learning?

## 3 Proposed Technique

In this section, we present our proposed algorithm, FOGZO, a first-order-guided zeroth-order gradient estimator. FOGZO leverages the insight that since the STE achieves good empirical results, its estimated gradient must be *mostly* accurate except for a few edge cases (Section A) where it may point in wrong directions. This makes the STE a good *source of bias* to reduce both the computation and variance of n-SPSA. In addition, FOGZO utilizes a key property of n-SPSA—that it suppresses bad gradient samples—to suppress the errors introduced by STE (Section 3.2) in these edge cases, achieving a method that both relies on and is better than the STE. We present the full FOGZO algorithm, provide a heuristic derivation, and finally discuss how to properly select hyperparameters for FOGZO.

### 3.1 Algorithm

FOGZO can be summarized in Equations 5 and 6. The pseudocode equivalent is presented in Algorithm 1. Instead of relying solely on the unbiased component $u_i$ in Equation 4, FOGZO (Equation 5) additionally introduces a biased component using the gradient estimated by the STE, denoted as $g$. While it is possible to use $g$ directly, we note that n-SPSA can benefit from using oracles with properties like unit magnitude, zero mean, and symmetry. As such, we first normalize $g$ to a unit vector $\hat{g}$ and then randomly choose between $\hat{g}$ and $-\hat{g}$ with a 50% chance each (denoted as $s_i\hat{g}$) to ensure the biased component is symmetric and zero-mean. In Equation 5, the biased component $s_i\hat{g}$ and unbiased component $u_i$ are mixed into a single sample denoted as $v_i$. We additionally introduce a mixing ratio $\beta$. As we will discuss in Section 3.2, $\beta$ determines the degree of trust in the STE. Similar to n-SPSA in Equation 4, finite difference is then used to generate the final FOGZO gradient estimation $G$ as shown in Equation 6. The mixing ratio $\beta$ is linearly decayed from 1 to some hyperparameter $\beta_{\min} < 1$, as shown in Algorithm 1, line 3, which we discuss more in the next section.

$$v_i = \sqrt{\beta}s_i\hat{g} + \sqrt{1 - \beta_i}u_i, \text{ where } s_i \sim 2 \cdot \text{Ber}(0.5) - 1, \hat{g} = \frac{g}{||g||}, \text{ and } u_i \sim p(u) \qquad (5)$$

$$G = \frac{1}{n}\sum_{i=1}^{n}\frac{\hat{L}(\theta + \epsilon v_i) - \hat{L}(\theta - \epsilon v_i)}{2\epsilon}v_i \qquad (6)$$

Notably, as shown in Algorithm 1, FOGZO estimates the gradient twice, using both first-order (one forward pass and one backward pass, Line 5) and zeroth-order ($2n$ forward passes, Lines 8 to 16). While this makes FOGZO more expensive than the STE, having a source of bias allows FOGZO to be orders of magnitude faster ($n \in O(1)$) than n-SPSA, while still retaining some of its benefits. In practice, we always use $n = 1$ for any deep models, which is equivalent to an additional computation cost of 2 forward passes compared to training with the STE. Similar to MeZO (31), we perturb and restore the parameters in place for memory efficiency (Lines 10, 12, and 14).

### 3.2 Heuristic Derivation

In this section, we perform heuristic derivation to understand the expected behaviour of the FOGZO gradient $G$. Utilizing the fact that $\hat{L}_{\text{smooth}}$ from Equation 3 is an approximation to $\hat{L}$ and first-order

---

**Algorithm 1** FOGZO Gradient Descent

---

1: **function** FOGZO($n$, $\beta_{\min}$, $\epsilon$, $p(u)$)
2:     **for** $t \leftarrow 0$ **to** $T - 1$ **do**
3:         $\beta = (1 - \frac{t}{T}) \cdot (1 - \beta_{\min}) + \beta_{\min}$ # linear beta decay from 1 to $\beta_{\min}$
4:         model.weight.grad = 0
5:         model(data).loss.backward() # regular backward pass using some pre-configured STE
6:         $g$ = model.weight.grad
7:         model.weight.grad = 0
8:         **for** $i \leftarrow 1$ **to** $n$ **do**
9:             Generate $v_i$ from Equation 5
10:            model.weight += $\epsilon v_i$ # perturb model weights in-place
11:            loss1 = model(data).loss # positive direction loss
12:            model.weight -= $2\epsilon v_i$ # perturb model weights in-place
13:            loss2 = model(data).loss # negative direction loss
14:            model.weight += $\epsilon v_i$ # restore model weights in-place
15:            model.weight.grad += (loss1 - loss2) / $(2n\epsilon)$ * $v_i$ # Equation 6
16:         **end for**
17:         optimizer.step() # gradient descent
18:     **end for**
19: **end function**

---

Taylor expansion, we derive Equation 7.

$$
\begin{aligned}
G &= \frac{1}{n}\sum_{i=1}^{n}\frac{\hat{L}(\theta + \epsilon v_i) - \hat{L}(\theta - \epsilon v_i)}{2\epsilon}v_i \approx \frac{1}{n}\sum_{i=1}^{n}\frac{\hat{L}_{\text{smooth}}(\theta + \epsilon v_i) - \hat{L}_{\text{smooth}}(\theta - \epsilon v_i)}{2\epsilon}v_i \\
&\approx \frac{1}{n}\sum_{i=1}^{n}\frac{\hat{L}_{\text{smooth}}(\theta) + (\epsilon v_i)^T\nabla\hat{L}_{\text{smooth}}(\theta) - \hat{L}_{\text{smooth}}(\theta) + (\epsilon v_i)^T\nabla\hat{L}_{\text{smooth}}(\theta)}{2\epsilon}v_i \\
&= \frac{1}{n}\sum_{i=1}^{n}(v_i^T\nabla\hat{L}_{\text{smooth}}(\theta))v_i = \frac{1}{n}\sum_{i=1}^{n}v_i(v_i^T\nabla\hat{L}_{\text{smooth}}(\theta)) = \frac{1}{n}\sum_{i=1}^{n}v_i v_i^T\nabla\hat{L}_{\text{smooth}}(\theta)
\end{aligned}
\tag{7}
$$

See Section B for a discussion on the approximation error of the two $\approx$ symbols in Equation 7.

Substituting Equation 5 into Equation 7 and using the fact that $u$ is zero-mean and unit-variance/isotropic, we then derive an approximation of $E[G]$ in Equation 8, which suggests with large $n$, the FOGZO gradient $G$ approximates a linear interpolation of the unbiased gradient $\nabla\hat{L}_{\text{smooth}}$ and its biased projection onto the rank-1 subspace described by the normalized STE gradient $\hat{g}$. This biased projection term can be interpreted as rank-1 gradient compression similar to MeZO (31). When $n$ is small, the unbiased term has a much larger variance than the biased term.

$$
\begin{aligned}
E[G] &\approx E[vv^T]\nabla\hat{L}_{\text{smooth}}(\theta) \\
&= E[(\sqrt{\beta}s\hat{g} + \sqrt{1-\beta}u)(\sqrt{\beta}s\hat{g} + \sqrt{1-\beta}u)^T]\nabla\hat{L}_{\text{smooth}}(\theta) \\
&= E[\beta s^2\hat{g}\hat{g}^T + (1-\beta)uu^T + \sqrt{\beta(1-\beta)}(s\hat{g}u^T + su\hat{g}^T)]\nabla\hat{L}_{\text{smooth}}(\theta) \\
&= (\beta\hat{g}\hat{g}^T + (1-\beta)I)\nabla\hat{L}_{\text{smooth}}(\theta) \\
&= \beta\underbrace{\hat{g}\hat{g}^T\nabla\hat{L}_{\text{smooth}}(\theta)}_{\text{biased}} + (1-\beta)\underbrace{\nabla\hat{L}_{\text{smooth}}(\theta)}_{\text{unbiased}}
\end{aligned}
\tag{8}
$$

Equation 8 suggests FOGZO can **suppress the flaws of the STE**. When the STE is accurate, or when it points in a direction similar to $\nabla\hat{L}_{\text{smooth}}$, $\hat{g}^T\nabla\hat{L}_{\text{smooth}}$ evaluates to a large scalar, and the biased term is allowed to contribute to $G$. However, when the STE is inaccurate, i.e. when it points in an orthogonal direction to $\nabla\hat{L}_{\text{smooth}}$, the scalar $\hat{g}^T\nabla\hat{L}_{\text{smooth}}$ evaluates to 0, and the biased term no longer contributes to $G$. In essence, the magnitude of the biased term depends on the correctness of the STE gradient, or whether it points in a similar direction to $\nabla\hat{L}_{\text{smooth}}$.

In addition, Equation 8 provides **insight on how to select the mixing ratio** $\beta$. When $\hat{g}$ aligns with $\nabla\hat{L}_{\text{smooth}}$, the STE gradient is more "trustworthy". As such, we can use a large $\beta$ to increase the

contributions of the biased term while reducing the unbiased term which has larger variance. However, when the STE misaligns with $\nabla \hat{L}_{\text{smooth}}$, the model is in an edge case where the STE gradient is inaccurate. In such cases, the biased term has a relatively small magnitude, and training can stop making progress. To allow training to proceed, we can decrease $\beta$ to increase the unbiased yet noisy term to prevent gradient signal from vanishing. Since the flawed behaviours of the STE tend to happen at the later stages of training (33), we decided to set $\beta = 1$ at the beginning of training, and gradually decay it to $\beta_{\text{min}}$ as shown in Algorithm 1. This coincides with the fact that at the later stages of training, the learning rate is much smaller, which makes the network more resistant to large gradient variance. Therefore, a small $\beta$ which increases the unbiased term, along with the gradient variance it introduces, is more acceptable at the later stages of training.

### 3.3 Selecting the Right Smoothness $\epsilon$ and Perturbation $p(u)$

Choosing appropriate smoothness $\epsilon$ and perturbation $p(u)$ is critical for FOGZO's performance, yet tuning them by grid search is prohibitively expensive. We therefore seek a principled way to configure these hyperparameters given a particular STE to be used with FOGZO. Our key insight is that every STE implicitly defines a form of smoothing. An STE estimates the Jacobian of a non-differentiable operator (round or sign) with the Jacobian of a surrogate function (smooth_round or smooth_sign), and the surrogate can be viewed as the expected value of the original operator under some random perturbation. In other words, each STE corresponds to a pair $(\bar{\epsilon}, \bar{p}(u))$ describing its implicit smoothness and perturbation distribution. If we can determine this implicit pair, we can use it to guide the choice of FOGZO's $(\epsilon, p(u))$. Doing so aligns FOGZO's randomized smoothing behavior with that of the STE, while leaving the granularity of smoothing (operation-wise versus full-model) as the only remaining difference. Formally, for an operator $h(x)$ (such as sign) and its STE surrogate $h_{\text{smooth}}(x)$, we assume:

$$h_{\text{smooth}}(x) = E_{u \sim \bar{p}(u)}[h(x + \bar{\epsilon}u)] \tag{9}$$

and solve for $(\bar{\epsilon}, \bar{p}(u))$. As an example, consider the hardtanh STE (38), which approximates the sign function. We express the hardtanh function, defined as $\text{hardtanh}(x) = \text{clip}(x, -1, 1)$, as the expected sign function under a perturbation distribution $\bar{p}(u)$ and solve for that $\bar{p}(u)$. For notational convenience, we denoted $\bar{P}$ as the CDF of $\bar{p}$, which suggests $\bar{P}(-\infty) = 0$ and $\bar{P}(\infty) = 1$.

$$
\begin{aligned}
\text{hardtanh}(x) &= E_{u \sim \bar{p}(u)}[\text{sign}(x + \bar{\epsilon}u)] \\
&= \int_{-\infty}^{\infty} \text{sign}(x + \bar{\epsilon}u)\bar{p}(u)du \qquad \text{(substitute } z = \bar{\epsilon}u \text{ and } du = dz/\bar{\epsilon}) \\
&= \int_{-\infty}^{\infty} \text{sign}(x + z)\bar{p}(z/\bar{\epsilon})/\bar{\epsilon}dz = \int_{-\infty}^{-x} -\bar{p}(z/\bar{\epsilon})/\bar{\epsilon}dz + \int_{-x}^{\infty} \bar{p}(z/\bar{\epsilon})/\bar{\epsilon}dz \\
&= -\bar{P}(-x/\bar{\epsilon}) + \bar{P}(-\infty) + \bar{P}(\infty) - \bar{P}(-x/\bar{\epsilon}) \\
&= 1 - 2\bar{P}(-x/\bar{\epsilon}) = \text{hardtanh}(x)
\end{aligned} \tag{10}
$$

Differentiating both sides with respect to $x$ gives the corresponding PDF $\bar{p}(u)$.

$$
\begin{aligned}
2\bar{p}(-x/\bar{\epsilon})/\bar{\epsilon} &= \text{hardtanh}'(x) \qquad \text{(substitute } y = -x/\bar{\epsilon} \text{ and } x = -\bar{\epsilon}y) \\
2\bar{p}(y)/\bar{\epsilon} &= \text{hardtanh}'(-\bar{\epsilon}y) = \text{hardtanh}'(\bar{\epsilon}y) \\
\bar{p}(y) &= \begin{cases} \bar{\epsilon}/2, & \text{if } -1/\bar{\epsilon} \leq y \leq 1/\bar{\epsilon} \\ 0, & \text{otherwise} \end{cases}
\end{aligned} \tag{11}
$$

Hence $p(u)$ corresponds to a uniform distribution $U(-1/\bar{\epsilon}, 1/\bar{\epsilon})$. To ensure $\bar{p}(u)$ is unit-variance/isotropic, we can set $\bar{\epsilon} = 1/\sqrt{3}$ and $\bar{p}(u) = U(-\sqrt{3}, \sqrt{3})$. Similarly, we solve for the $\bar{\epsilon}$ and $\bar{p}(u)$ for the identity-STE(3), the tanh-STE (38), the ApproxSign-STE (26) and present our results in Equation 12. See Section D for proofs and further discussion.

$$I(x) = E_{u \sim \bar{p}(u)}[\text{round}(x + \bar{\epsilon}u)], \text{ where } \bar{\epsilon} = 1/(2\sqrt{3}) \text{ and } \bar{p}(u) = U(-\sqrt{3}, \sqrt{3})$$

$$\tanh(x) = E_{u \sim \bar{p}(u)}[\text{sign}(x + \bar{\epsilon}u)], \text{ where } \bar{\epsilon} = \pi/\sqrt{12} \text{ and } \bar{p}(u) = \bar{\epsilon}(1 - \tanh^2(\bar{\epsilon}u))/2$$

$$\text{ApproxSign}(x) = E_{u \sim \bar{p}(u)}[\text{sign}(x + \bar{\epsilon}u)], \text{ where } \bar{\epsilon} = 1/\sqrt{6} \text{ and } \bar{p}(u) = \text{tri}(u/\sqrt{6})/\sqrt{6}$$

$$\tag{12}$$

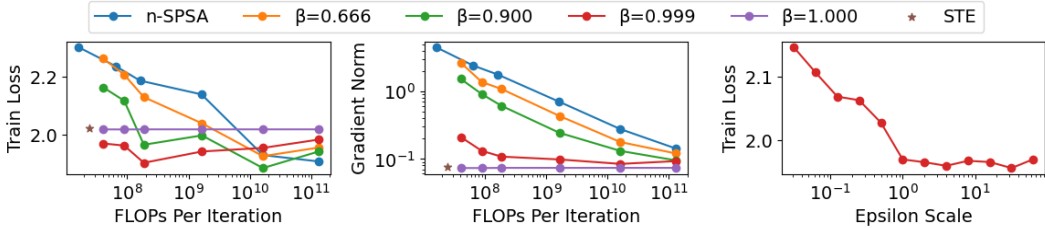

Figure 1: Comparing the STE, n-SPSA, and FOGZO on a 2-layer MLP with 2-bit weights.

If the round or sign function is directly applied on the raw parameter $\theta$, we set $\epsilon = \bar{\epsilon}$ and $p(u) = \bar{p}(u)$. For a more general quantization setup like Equation 1, where $\theta$ is first processed by $f^{-1}$, i.e. divided by a quantization scale $\alpha$ and then discretized, we set $\epsilon = \alpha\bar{\epsilon}$ and $p(u) = \bar{p}(u)$. If there are multiple quantization scales in a model, we use the averaged quantization scale as $\alpha$.

## 4  Evaluation

We first compare the STE, n-SPSA, and FOGZO with experiments on shallow networks in Section 4.1, as running n-SPSA with large $n$ is infeasible on deeper networks. In Section 4.2, we compare the STE and FOGZO on deeper networks with $n = 1$. Lastly, in Section 4.3, we compare the evaluation quality of FOGZO against the STE under the same training time.

### 4.1  Shallow Networks: Comparing the STE, n-SPSA, and FOGZO

In this section, we seek to confirm the following 3 statements: the ZO method n-SPSA can outperform the STE when there is sufficient computation (large sample size $n$); when there is insufficient computation, n-SPSA can underperform by a large margin due to its large gradient variance; FOGZO can significantly reduce this reliance on computation, and can outperform the STE with only a small increase in computation budget.

We train MNIST on a 2-layer MLP (784-10-10) with the identity STE, the ZO method n-SPSA, and FOGZO. We quantize weights to 2-bit with a constant quantization scale $\alpha$ that is shared by both linear layers. Followed from the insight in Section 3.3, we use $\epsilon = \alpha\bar{\epsilon} = \alpha/(2\sqrt{3})$ and $p(u) = U(-\sqrt{3}, \sqrt{3})$. For FOGZO, we use $\beta \in \{0.666, 0.9, 0.999, 1\}$. For both FOGZO and n-SPSA, we use $n \in \{1, 4, 10, 100, 1000, 7960\}$. We do not decay $\beta$ for simplicity and simply keep it as a constant hyperparameter. We present our results in Figure 1. For more hyperparameters, see Section E.1.

As shown in Figure 1 left, while n-SPSA can outperform the identity STE, it comes at the cost of requiring orders of magnitude more computation. When n-SPSA has a low $n$, it underperforms due to the large gradient variance (shown in Figure 1 middle). FOGZO addresses this limitation. **As $\beta$ increases, FOGZO consistently reduces gradient norm/variance and training loss for low computation.** At $\beta = 0.999$ and $n = 1$, FOGZO can outperform the STE, while requiring only 2 additional forward passes per iteration. Crucially, when $\beta = 1$, FOGZO completely relies on its source of bias, and therefore cannot outperform the STE. This suggests **the optimal $\beta$ is close to, but not exactly, one.**

In Figure 1 right, we additionally test the insights in Section 3.3 by setting $\epsilon = c\alpha\bar{\epsilon}$ and sweeping a range of epsilon scale $c$, while keeping $n = 1$ and $\beta = 0.999$. Results suggest that $c = 1$ is one optimal configuration, although larger values of $c$ are equally good.

The experiments on shallow networks suggest FOGZO may be a promising alternative for the STE. At the cost of 2 additional forward passes per iteration, FOGZO can outperform the STE. This is especially useful for cases where a mild increase in training cost is acceptable as long as inference quality can be maximized. However, can FOGZO outperform the STE in deeper networks?

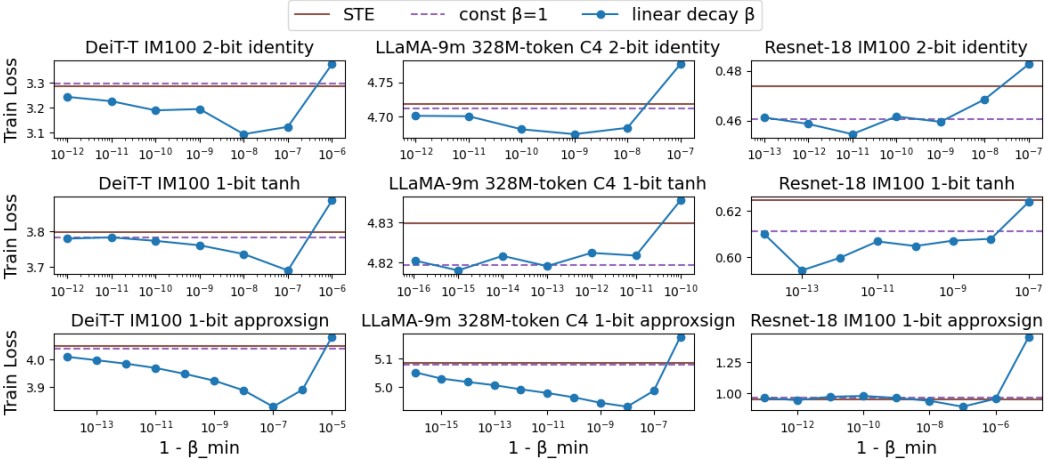

Figure 2: Training Loss vs $\beta_{\min}$. For 2-bit weights, we compare Identity-STE against Identity-FOGZO. For 1-bit weights, we compare tanh against tanh-FOGZO and ApproxSign against ApproxSign-FOGZO.

| Model | Dataset | Identity-STE | Identity-FOGZO | tanh | tanh-FOGZO | ApproxSign | ApproxSign-FOGZO |
|---|---|---|---|---|---|---|---|
| DeiT-Tiny | Imagenet-100 | 62.72 | **70.06** | 41.98 | **46.8** | 30.16 | **39.61** |
| LLaMA-9m | 328M C4 tokens | 109.95 | **105.64** | 123.97 | **121.51** | 159.17 | **137** |
| Resnet18 | Imagenet-100 | 79.92 | **80.42** | 74.68 | **75.02** | 68.62 | **70.91** |

Table 1: Accuracy or perplexity of models with 1-bit or 2-bit weights and fixed $\alpha$.

## 4.2 Deeper Networks: Comparing the STE and FOGZO

In this section, we confirm that in deeper networks, FOGZO can still outperform a variety of STEs with a slight increase in computation budget. We select a list of representative Deep Learning model architectures: Resnet (18), DeiT (10; 48), and LLaMA (49). Due to resource constraints, we train on the smaller variants of these architectures: Resnet-18, Resnet-50, ViT-Tiny, ViT-Small, and LLaMA-{9M, 20M, 30M, 50M, 100M, 200M}. We evaluate the most challenging QAT precisions: 2-bit and 1-bit. For datasets, we use Imagenet-1K (41), Imagenet-100(7), and C4 (39). For FOGZO, since it is impossible to use a large $n$, we simply fix $n = 1$. Unlike Section 4.1, here we use linearly scheduled $\beta$s with the hyperparameter $\beta_{\min}$. Similarly, we set $\epsilon$ and $p(u)$ based on the insight in Section 3.3.

### 4.2.1 FOGZO can Improve a Variety of STEs

In this section, we confirm that **regardless of the STE used as a source of bias, FOGZO can improve the gradient quality**. We compare identity-STE against identity-FOGZO for 2-bit quantization, tanh-STE against tanh-FOGZO and ApproxSign-STE against ApproxSign-FOGZO for binary quantization. Similar to Section 4.1, for every linear and convolution layer, we inject a weight quantizer, where the quantization step size $\alpha$ is fixed and shared by all layers in a model. For more hyperparameters, see Section E.2.1.

We perform a $\beta_{\min}$-sweep and present the results in Figure 2. As shown, regardless of model architecture, dataset, and the chosen source of bias, FOGZO (blue) is capable of outperforming the STE (brown) within a range of $\beta_{\min} \approx 1$ (or $1 - \beta_{\min} \approx 0$). As $\beta_{\min}$ increases, training loss rapidly reduces. However, at $\beta_{\min} \approx 1$ the benefit of FOGZO saturates as training becomes completely dominated by STE-biased signal. At $\beta_{\min} = 1$ (labeled as "const $\beta = 1$") FOGZO cannot outperform the STE consistently. For each graph in Figure 2, we select the $\beta_{\min}$ with the lowest training loss and report its evaluation quality in Table 1. For vision models, we report top-1 accuracy. For language models, we report perplexity. As shown, with a properly tuned $\beta_{\min}$, FOGZO can outperform the STE by a large margin.

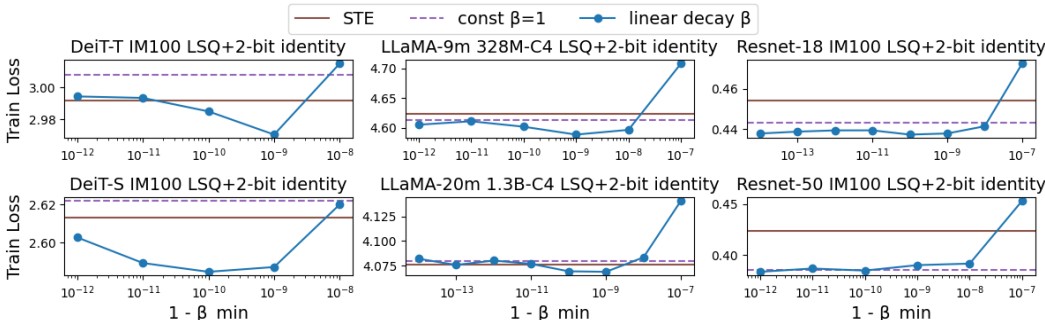

Figure 3: Training Loss vs $\beta_{\min}$. We compare LSQ+Identity-STE against LSQ+Identity-FOGZO.

| Model | Dataset | LSQ + Identity-STE | LSQ + Identity-FOGZO |
|---|---|---|---|
| DeiT-Tiny | Imagenet-100 | $3\pm0.017$ / $72.76\pm0.62$ | $\mathbf{2.98}\pm0.012$ / $\mathbf{72.92}\pm0.60$ |
| DeiT-Small | Imagenet-100 | $2.62\pm0.012$ / $79.55\pm0.86$ | $\mathbf{2.57}\pm0.016$ / $\mathbf{80.06}\pm0.48$ |
| DeiT-Tiny | Imagenet-1K | $4.26\pm0.015$ / $63.19\pm0.30$ | $\mathbf{4.25}\pm0.025$ / $\mathbf{63.38}\pm0.52$ |
| LLaMA-9m | 328M C4 tokens | $4.66\pm0.060$ / $104.31\pm6.54$ | $\mathbf{4.62}\pm0.040$ / $\mathbf{99.74}\pm4.42$ |
| LLaMA-9m | 3.28B C4 tokens | $4.41\pm0.020$ / $82.54\pm1.72$ | $\mathbf{4.38}\pm0.020$ / $\mathbf{79.94}\pm1.18$ |
| LLaMA-20m | 1.3B C4 tokens | $4.084\pm0.020$ / $59.21\pm1.22$ | $\mathbf{4.079}\pm0.020$ / $\mathbf{58.93}\pm0.90$ |
| LLaMA-20m | 13B C4 tokens | $3.94\pm0.008$ / $50.85\pm0.76$ | $\mathbf{3.93}\pm0.008$ / $\mathbf{50.61}\pm0.64$ |
| Resnet-18 | Imagenet-100 | $0.45\pm0.0084$ / $\mathbf{80.23}\pm0.84$ | $\mathbf{0.44}\pm0.0098$ / $80.04\pm1.14$ |
| Resnet-50 | Imagenet-100 | $0.43\pm0.0182$ / $82.81\pm0.82$ | $\mathbf{0.39}\pm0.0086$ / $\mathbf{83.67}\pm0.26$ |
| Resnet-18 | Imagenet-1K | $1.493\pm0.0032$ / $\mathbf{66.51}\pm0.30$ | $\mathbf{1.487}\pm0.0108$ / $66.13\pm0.50$ |

Table 2: Cross-entropy loss / accuracy or perplexity of models with 2-bit weights and learnable $\alpha$.

### 4.2.2 Integration with SOTA Methods and Scaling up

**Weight-Only Quantization:** We test FOGZO's compatibility with a SOTA QAT method LSQ (13). In Section 4.2.1, the quantization setup is rather naive as the quantization scale $\alpha$ is predetermined and shared by all layers. Here, we make $\alpha$ learnable and unique per layer and use the identity STE for 2-bit weight quantization. We follow the insight from Section 3.3, and use the average quantization scale $\alpha$ to set the smoothness. In addition, we further train models $2\times$ to $4\times$ larger than the ones in Section 4.2.1 to see if FOGZO scales with parameter count. Lastly, we increase our dataset size by a factor of $10\times$ and test FOGZO's extensibility to larger datasets. This is done by switching from Imagenet-100 to Imagenet-1K, and increasing the number of C4 tokens by $10\times$. We report in Figure 3 and Table 2 the training loss and evaluation accuracy or perplexity of the STE and FOGZO with the optimal $\beta_{\min}$. To ensure consistency, in each cell in Table 2, we report the mean and two standard deviation of five different runs. **Despite the change in quantizer design and model/dataset size, FOGZO is still capable of outperforming the STE**.

**Weight-Activation Quantization:** We further evaluate the effectiveness of FOGZO on SOTA weight-activation quantization methods QuEST (35) and LSQ. We use the publicly available source code of QuEST (35) and added the implementation of Algorithm 1, and pre-train LLaMA models with $n$ non-embedding parameters plus $e$ embedding parameters, where $n \in \{30M, 50M, 100M, 200M\}$ and $e \in \{65M, 75M, 100M, 100M\}$. For all experiments, we use a $\beta_{\min}$ of 1 - 1e-10, quantize activation and weights of all linear layers to 2-bit, and pre-train with $100n$ C4 tokens. We present our results in Table 3. Results suggest **FOGZO can improve both QuEST and LSQ when activations and weights are quantized**, and can even shrink the perplexity difference between these methods.

### 4.3 Reducing Training Time Overhead

In this section, we show the training time overhead of FOGZO can be drastically reduced by augmenting Algorithm 1 to perform the first $r\%$ of training in STE mode, allowing FOGZO to achieve **lower perplexity than the STE under the same training time**. Concretely, we only execute lines 6 to 16 in Algorithm 1 when $t > r/100 * T$. Note that $r = 0$ corresponds to the unmodified

| | | | | | | |
|---|---|---|---|---|---|---|
| Total Parameters $n+e$ | 95M | 95M | 95M | 95M | 125M | 125M |
| $n$ | 30M | 30M | 30M | 30M | 50M | 50M |
| Tokens | 3B | 3B | 3B | 3B | 5B | 5B |
| Method | QuEST | QuEST-FOGZO | LSQ | LSQ-FOGZO | QuEST | QuEST-FOGZO |
| Perplexity | 37.75 | **37.37** | 39.06 | **37.38** | 32.28 | **32.06** |
| Total Parameters $n+e$ | 125M | 125M | 200M | 200M | 300M | 300M |
| $n$ | 50M | 50M | 100M | 100M | 200M | 200M |
| Tokens | 5B | 5B | 10B | 10B | 20B | 20B |
| Method | LSQ | LSQ-FOGZO | QuEST | QuEST-FOGZO | QuEST | QuEST-FOGZO |
| Perplexity | 33.28 | **32.41** | 26.63 | **26.45** | 22.90 | **22.72** |

Table 3: Perplexity of LLaMA models trained using LSQ, LSQ-FOGZO, QuEST, and QuEST-FOGZO with 2-bit weights and 2-bit activations.

| Method | C4 Tokens (billion) | Evaluation Perplexity | Training Time (hours) |
|---|---|---|---|
| STE | 3.174 | 38.69 | 3.3 |
| 90% STE + 10% FOGZO | 3 | **38.67** | 3.3 |
| STE | 3.348 | 38.43 | 3.5 |
| 80% STE + 20% FOGZO | 3 | **38.38** | 3.5 |
| STE | 3.522 | 38.25 | 3.7 |
| 70% STE + 30% FOGZO | 3 | **37.93** | 3.7 |

Table 4: Training Time vs. Perplexity of LLaMA-30m on Nvidia RTX 5090

Algorithm 1. We use LSQ-STE and LSQ-FOGZO to train a LLaMA model with 30M non-embedding parameters on one Nvidia RTX 5090 GPU and report the training time and C4 evaluation perplexity in Table 4. For a fair comparison, we artificially increase the training time of the LSQ-STE runs by adding more pre-training tokens, while decreasing the training time of the LSQ-FOGZO runs by choosing $r \in \{70, 80, 90\}$. Results suggest FOGZO can outperform the STE when training time is identical. In addition, FOGZO can have higher data efficiency than STE, as **FOGZO runs use less training data to achieve lower perplexity**.

## 5 Related Works and Limitations

Most existing QAT methods use Backpropagation (40) to calculate gradient estimates, including but are not limited to the STE/operation-wise Jacobian estimators (3; 26; 35; 50), Coordinate Descent (30), Stochastic Rounding (17), Pseudo-Quantization Noise (44), and EWGS (22). FOGZO is orthogonal to these works and may benefit from using them as a source of bias. Existing zeroth-order methods(54; 31; 6) use only forward passes to estimate gradients, and are either specialized for fine-tuning, or underperforms compared to first-order methods. FOGZO is specialized for pre-training, and is the first partially-zeroth-order method to consistently outperform a first-order method STE for DNNs.

Although FOGZO works well with standard mixed-precision training methods (32) that keep weights in 32-bit, FOGZO suffers from loss of precision when weights are stored in 16-bit due to the challenge of modifying weights with small updates. Because of this same challenge, SOTA LLM pre-training pipelines like DeepSeek-V3 (8) still use 32-bit weights. We expect that as the field evolves, newer techniques will be developed to reduce rounding error, enabling large-scale pre-training with BF16 weights. These newer techniques can then be used to relieve the numerical issues of FOGZO.

## 6 Conclusion

In this work, we compare two QAT methods, the STE and n-SPSA. We introduce FOGZO which utilizes the STE as a source of bias to accelerate n-SPSA, creating the method FOGZO. We show that FOGZO can suppress the flaws of its bias source, allowing FOGZO to improve upon the STE. Experiments shows FOGZO can improve the gradient quality of a variety of STEs and for different model architectures and datasets. We hope FOGZO can influence future research in STEs as FOGZO provides a quantitative measure of STE gradient quality.

## Acknowledgements

This work was supported by the Natural Sciences and Engineering Research Council of Canada (NSERC) through a Discovery Grant and an NSERC Strategic Network Grant.

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

# A The STE is Occasionally Flawed

In this section we demonstrate, with examples, that the STE can occasionally produce gradients with wrong directions.

Suppose we wish to optimize the objective function $h(\theta) = g(\text{round}(\theta))$, where $\theta \in \mathbb{R}$ and $g(\psi) = \psi^3 - \psi/4$. Note $h(\theta)$ is a non-decreasing function (visualized in Figure 4) as it satisfies the property $h(\theta - \epsilon) \leq h(\theta)$ for any $\theta$ and any $\epsilon > 0$. In other words, $h$ can always be made smaller by reducing $\theta$, regardless of what $\theta$ is.

This suggests valid gradient estimators must encourage $\theta$ to reduce regardless of its value, or any high-quality gradient estimator must produce a positive gradient.

Suppose we use the identity STE and its estimated gradient:

$$h'_{\text{ste}}(\theta) = g'(\text{round}(\theta)) = 3(\text{round}(\theta))^2 - 1/4 \tag{13}$$

The STE gradient $h'_{\text{ste}}(\theta)$ is **not** always positive. When $-0.5 \leq \theta < 0.5$, we have $h'_{\text{ste}}(\theta) < 0$, which suggests we should **increase** $\theta$. This means any $\theta$ initialized between -0.5 and 0.5, will gradually increase until it reaches or becomes larger than $0.5$. When $\theta \geq 0.5$, $h'_{\text{ste}}(\theta)$ becomes positive again, and the gradient descent algorithm will decrease $\theta$ until it becomes less than $0.5$. This cycle repeats itself, causing $\theta$ to always oscillate around $0.5$. In deep neural networks, the problem of parameter oscillation has been shown to slow down convergence when training with the STE (25; 33).

However, we emphasize this is an **occasional flaw**, or an **edge case** of the STE gradient. Indeed, $h'_{\text{ste}}(\theta)$ is almost always positive, which suggest we should almost always reduce $\theta$. Even when the STE gradient makes mistakes, complementary techniques like SGD with momentum can help escape the local minima at $-0.5 \leq \theta < 0.5$. However, this is not a guarantee, and the effects of momentum depends on hyperparameters like the learning rate. Using the STE gradient requires further tuning of these hyperparameters, which can be undesired.

We next show that this is not a flaw of just the **identity** STE. Suppose we use soft rounding (2), we have the estimated gradient

$$
\begin{aligned}
h'_{\text{soft\_rounding}}(\theta) &= g'(\text{round}(\theta)) \cdot \text{soft\_rounding}'(\theta) \\
&= (3(\text{round}(\theta))^2 - 1/4) \cdot \text{soft\_rounding}'(\theta)
\end{aligned}
\tag{14}
$$

Since soft_rounding is an increasing function, $\text{soft\_rounding}'(\theta)$ is always positive. Therefore, $h'_{\text{soft\_rounding}}(\theta)$ is still negative at $-0.5 \leq \theta < 0.5$, similar to $h'_{\text{ste}}(\theta)$. Therefore, the same oscillation behaviour will still happen.

The reason parameter oscillations happen, is because $h$ is fundamentally different from $g$ (visualized in Figure 4). The former is a non-decreasing function, whereas the latter is not. In essence, the additional round function hides the local minima of $g$, making $h$ a non-decreasing function.

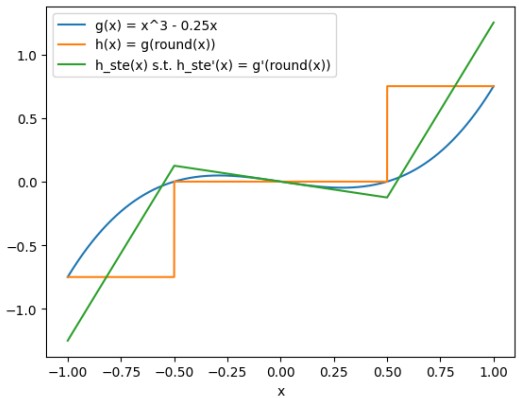

Figure 4: Example of the occasional flaw of the STE. $g(x) = x^3$, $h(x) = g(\text{round}(x))$, and $h'_{\text{ste}}(x) = 3(\text{round}(x))^2 - 1/4$.

Is our example objective function $h$ just a special counterexample, or does the same behaviour happen in deeper networks, where both $g$ and $h$ can be more complex? The answer is likely yes. **Quantization tends to hide local details, whereas the STE uses local details as a proxy to global behaviour**.

## B    Approximation Errors of the Heuristic Derivation

In Equation 7, we replace the non-differentiable $\hat{L}$ with a differentiable approximation $\hat{L}_{\text{smooth}}$, which enables a second-order Taylor Expansion that provides us some insight into the expected behaviour of the FOGZO gradient $G$. In this section, we quantify the approximation error of replacing $\hat{L}$ with $\hat{L}_{\text{smooth}}$ and the approximation error of using the first-order Taylor Expansion (i.e. dropping the second-order term, third-order term, etc.)

We note $\hat{L}_{\text{smooth}}$ has its own smoothness $\epsilon$ and perturbation distribution $p(u)$, not to be confused with the FOGZO-specific smoothness $\epsilon$ and perturbation distribution $p(u)$ in Equation 7. For clarification, we will refer to the smoothness and perturbation of $\hat{L}_{\text{smooth}}$ as $\hat{\epsilon}$ and $\hat{p}(u)$. We will choose $\hat{p}(u)$ to be uniformly distributed on the $d$-dimensional unit sphere. We assume $\hat{L}$ to be $(C_1, C_2)$ coarse Lipschitz, or that $|\hat{L}(x) - \hat{L}(y)| \leq C_1||x - y|| + C_2$.

The first approximation error $|\hat{L}(\theta) - \hat{L}_{\text{smooth}}(\theta)|$ can be bounded as follows.

$$
\begin{aligned}
|\hat{L}(\theta) - \hat{L}_{\text{smooth}}(\theta)| = |E[\hat{L}(\theta) - \hat{L}(\theta + \hat{\epsilon}u)]| \\
\leq E[|\hat{L}(\theta) - \hat{L}(\theta + \hat{\epsilon}u)|] \\
\leq C_1\hat{\epsilon}E[||\hat{\epsilon}u||] + C_2 \\
= C_1\hat{\epsilon} + C_2
\end{aligned}
\tag{15}
$$

For the second $\approx$ symbol, the approximation error can be quantified with a big-O bound.

Since

$$
\hat{L}_{\text{smooth}}(\theta + \epsilon v_i) = \hat{L}_{\text{smooth}}(\theta) + (\epsilon v_i)^T\nabla\hat{L}_{\text{smooth}}(\theta) + (\epsilon v_i)^T\nabla^2\hat{L}_{\text{smooth}}(\theta)(\epsilon v_i) + O(||\epsilon v_i||^3) \tag{16}
$$

and

$$
\hat{L}_{\text{smooth}}(\theta - \epsilon v_i) = \hat{L}_{\text{smooth}}(\theta) - (\epsilon v_i)^T\nabla\hat{L}_{\text{smooth}}(\theta) + (\epsilon v_i)^T\nabla^2\hat{L}_{\text{smooth}}(\theta)(\epsilon v_i) + O(||\epsilon v_i||^3) \tag{17}
$$

Subtracting the two, we have

$$
\hat{L}_{\text{smooth}}(\theta + \epsilon v_i) - \hat{L}_{\text{smooth}}(\theta - \epsilon v_i) = 2(\epsilon v_i)^T\nabla\hat{L}_{\text{smooth}}(\theta) + O(||\epsilon v_i||^3) \tag{18}
$$

Note that all even-order terms cancel out due to the symmetric nature of the central finite difference, leaving only the third-order term which is dominating for a sufficiently small $\epsilon$ and a Hessian-Lipschitz $\hat{L}_{\text{smooth}}$.

Next, we divide by $2\epsilon$ and multiply by $v_i$, and get

$$
\frac{\hat{L}_{\text{smooth}}(\theta + \epsilon v_i) - \hat{L}_{\text{smooth}}(\theta - \epsilon v_i)}{2\epsilon}v_i = v_iv_i^T\nabla\hat{L}_{\text{smooth}}(\theta) + O(\epsilon^2||v_i||^3)v_i \tag{19}
$$

where the term $O(\epsilon^2||v_i||^3)v_i$ is the approximation error of the second $\approx$ symbol.

## C    Random Sign Flip

As discussed in Section 3.1, FOGZO introduces a random variable $s_i$ that ensures $v_i = \sqrt{\beta}s_i\hat{g} + \sqrt{1-\beta}u_i$ is symmetric. Since the central finite difference method already introduces symmetry, the second-order term cancels out as discussed in Section B. Therefore, $s_i$ is not necessary in theory. However, in practice, we find using $s_i$ can improve the performance slightly. For example, when we train Resnet-50 on Imagenet-100, we notice that FOGZO (with $s_i$) achieves a training loss of 0.3872 (averaged over 5 runs), whereas a variant of FOGZO without $s_i$ achieves a training loss of 0.3883 (averaged over 5 runs). We attribute this to floating point error breaking the symmetry of the central finite difference method. Therefore, we include $s_i$ in FOGZO for its empirical performance gain.

# D   Operation-Wise Jacobian Estimators

In this section, we prove several rounding/sign gradient estimators, also known as STEs, are expressible as randomized smoothing variants of round/sign. We denote round as $R$ and sign as $S$. A smooth approximation of some function $h$ with smoothness $\epsilon$ and perturbation $p(u)$ is denoted as $h_{\epsilon,p(u)}$. In the main text, $h_{\epsilon,p(u)}$ is simply referred to as $h_{\text{smooth}}$.

## D.1   The Identity Function

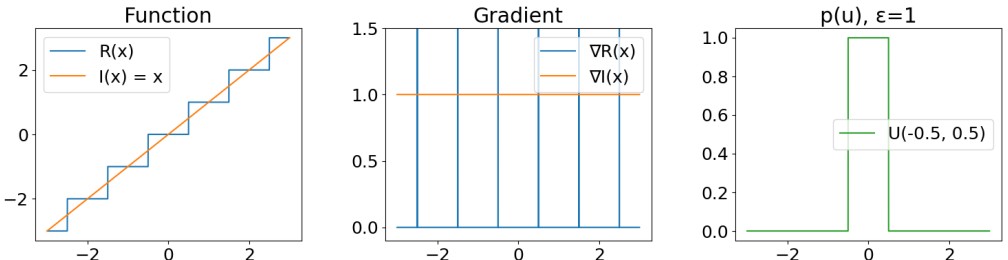

Figure 5: The identity STE uses the identity function to approximate rounding.

The identity STE (3) uses the identity function $I(x) = x$, visualized in Figure 5.

Statement to prove: $I(x) = R_{\epsilon,p(u)}(x)$, where $p(u) = U(-0.5/\epsilon, 0.5/\epsilon)$.

$$R_{\epsilon,U(-0.5/\epsilon,0.5/\epsilon)}(x)$$
$$= E_{u\sim U(-0.5/\epsilon,0.5/\epsilon)}[R(x+\epsilon u)]$$
$$= \int_{u=-0.5/\epsilon}^{0.5/\epsilon} R(x+\epsilon u)p(u)du \tag{20}$$
$$= \int_{u=-0.5/\epsilon}^{0.5/\epsilon} R(x+\epsilon u)\epsilon du$$

We substitute $z = x + \epsilon u$, $u = (z-x)/\epsilon$, and $du = dz/\epsilon$

$$R_{\epsilon,U(-0.5/\epsilon,0.5/\epsilon)}(x)$$
$$= \int_{z=x-0.5}^{x+0.5} R(z)dz$$
$$= \int_{z=x-0.5}^{\lfloor x\rfloor+0.5} R(z)dz + \int_{z=\lfloor x\rfloor+0.5}^{x+0.5} R(z)dz$$
$$= \int_{z=x-0.5}^{\lfloor x\rfloor+0.5} \lfloor x\rfloor dz + \int_{z=\lfloor x\rfloor+0.5}^{x+0.5} \lceil x\rceil dz \tag{21}$$
$$= \lfloor x\rfloor(\lfloor x\rfloor + 0.5 - (x-0.5)) + \lceil x\rceil(x+0.5 - (\lfloor x\rfloor + 0.5))$$
$$= \lfloor x\rfloor(\lfloor x\rfloor - x + 1)) + \lceil x\rceil(x - \lfloor x\rfloor)$$
$$= \lfloor x\rfloor(\lfloor x\rfloor - x)) + \lceil x\rceil(x - \lfloor x\rfloor) + \lfloor x\rfloor$$
$$= (-\lfloor x\rfloor)(x - \lfloor x\rfloor)) + \lceil x\rceil(x - \lfloor x\rfloor) + \lfloor x\rfloor$$
$$= (\lceil x\rceil - \lfloor x\rfloor)(x - \lfloor x\rfloor) + \lfloor x\rfloor$$
$$= (x - \lfloor x\rfloor) + \lfloor x\rfloor$$
$$= x \ \square$$

To ensure $p(u)$ has unit variance, we can use $\epsilon = 0.5/\sqrt{3}$ and $p(u) = U(-\sqrt{3}, \sqrt{3})$.

## D.2 Hardtanh

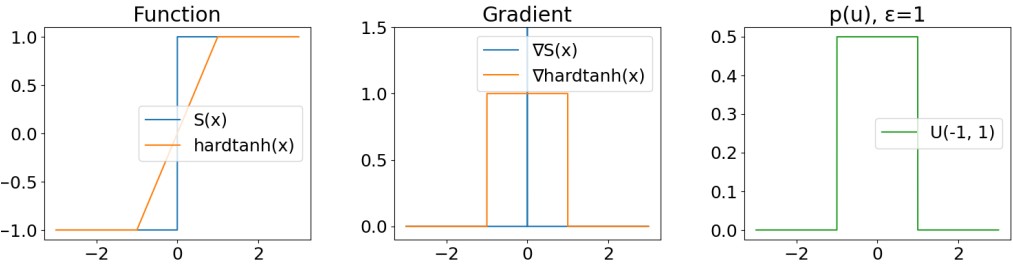

Figure 6: The Hardtanh function approximates the sign function.

The Hardtanh STE (38) uses the hardtanh function defined as follows:

$$\text{hardtanh}(x) = \begin{cases} -1, & \text{if } x < -1, \\ 1, & \text{if } x > 1, \\ x, & \text{otherwise} \end{cases} \tag{22}$$

Statement to prove: $\text{hardtanh}(x) = S_{\epsilon, p(u)}(x)$, where $p(u) = U(-1/\epsilon, 1/\epsilon)$.

$$
\begin{aligned}
S_{\epsilon=1,U(-1/\epsilon,1/\epsilon)}(x) \\
&= E_{u \sim U(-1/\epsilon,1/\epsilon)}[S(x + \epsilon u)] \\
&= \int_{u=-1/\epsilon}^{1/\epsilon} S(x + \epsilon u)p(u)du \\
&= \frac{\epsilon}{2} \int_{u=-1/\epsilon}^{1/\epsilon} S(x + \epsilon u)du
\end{aligned}
\tag{23}
$$

We substitute $z = \epsilon u$, $u = z/\epsilon$, and $du = dz/\epsilon$.

$$S_{\epsilon=1,U(-1/\epsilon,1/\epsilon)}(x) = \frac{1}{2} \int_{z=-1}^{1} S(x + z)dz \tag{24}$$

- We first consider the case $-1 \leq x \leq 1$.

$$
\begin{aligned}
S_{\epsilon=1,U(-1/\epsilon,1/\epsilon)}(x) \\
&= \frac{1}{2} \int_{-1}^{1} S(x + u)du \\
&= \frac{1}{2} \int_{-1}^{-x} S(x + u)du + \frac{1}{2} \int_{-x}^{1} S(x + u)du \\
&= -\frac{1}{2} \int_{-1}^{-x} 1du + \frac{1}{2} \int_{-x}^{1} 1du \\
&= -\frac{1}{2}((-x) - (-1)) + \frac{1}{2}(1 + x) \\
&= \frac{1}{2}(x - 1) + \frac{1}{2}(1 + x) \\
&= \frac{1}{2}(x - 1 + 1 + x) \\
&= \frac{1}{2}(2x) \\
&= x
\end{aligned}
\tag{25}
$$

- Then we consider the case $x < -1$.

$$S_{\epsilon=1, U(-1/\epsilon, 1/\epsilon)}(x)$$

$$= \frac{1}{2} \int_{-1}^{1} S(x+u)du$$

$$= -\frac{1}{2} \int_{-1}^{1} 1 du \qquad (26)$$

$$= -\frac{1}{2}(1-(-1))$$

$$= -\frac{1}{2} 2$$

$$= -1$$

- Finally, we consider the case $x > 1$.

$$S_{\epsilon=1, U(-1/\epsilon, 1/\epsilon)}(x)$$

$$= \frac{1}{2} \int_{-1}^{1} S(x+u)du$$

$$= \frac{1}{2} \int_{-1}^{1} 1 du \qquad (27)$$

$$= \frac{1}{2}(1-(-1))$$

$$= \frac{1}{2} 2$$

$$= 1 \ \square$$

To ensure $p(u)$ has unit-variance, we can use $\epsilon = 1/\sqrt{3}$ and $p(u) = U(-\sqrt{3}, \sqrt{3})$.

### D.3 Tanh-Based Functions

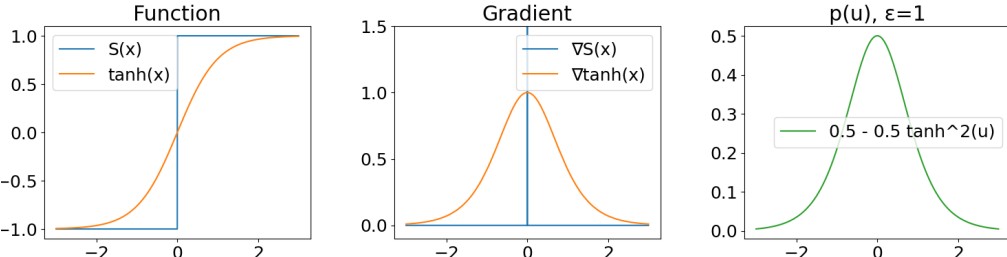

Figure 7: The tanh function approximates the sign function.

The tanh STE (38) uses the tanh function, which corresponds to smoothing the sign function via a perturbation distribution with PDF $p(u) = \frac{\epsilon}{2}(1 - \tanh^2(\epsilon u))$ and CDF $P(u) = \frac{1}{2} + \frac{1}{2}\tanh(\epsilon u)$. This distribution is equivalent to a logistic distribution with mean 0 and scale $\frac{1}{2\epsilon}$.

Statement to prove: $\tanh(x) = S_{\epsilon, p(u)}(x)$, where $p(u) = \frac{\epsilon}{2}(1 - \tanh^2(\epsilon u))$.

$$S_{\epsilon=1, p(u)}(x)$$

$$= E_{u \sim p(u)}[S(x + \epsilon u)]$$

$$= \int_{-\infty}^{\infty} S(x + \epsilon u) p(u) du$$

$$= \int_{-\infty}^{-x/\epsilon} S(x + \epsilon u) p(u) du + \int_{-x/\epsilon}^{\infty} S(x + \epsilon u) p(u) du$$

$$= \int_{-\infty}^{-x/\epsilon} -p(u) du + \int_{-x/\epsilon}^{\infty} p(u) du$$

$$= -\frac{1}{2} \int_{-\infty}^{-x/\epsilon} \epsilon(1 - \tanh^2(\epsilon u)) du + \frac{1}{2} \int_{-x/\epsilon}^{\infty} \epsilon(1 - \tanh^2(\epsilon u)) du \qquad (28)$$

$$= -\frac{1}{2} [\tanh(\epsilon u)]_{-\infty}^{-x/\epsilon} + \frac{1}{2} [\tanh(\epsilon u)]_{-x/\epsilon}^{\infty}$$

$$= -\frac{1}{2} [\tanh(-x) - \tanh(-\infty)] + \frac{1}{2} [\tanh(\infty) - \tanh(-x)]$$

$$= -\frac{1}{2}\tanh(-x) + \frac{1}{2}\tanh(-\infty) + \frac{1}{2}\tanh(\infty) - \frac{1}{2}\tanh(-x)$$

$$= -\frac{1}{2}\tanh(-x) - \frac{1}{2}\tanh(-x)$$

$$= -\tanh(-x)$$

$$= \tanh(x) \ \square$$

To ensure $p(u)$ has unit variance, we use $\epsilon = \frac{\pi}{\sqrt{12}}$ and $p(u) = \frac{\epsilon}{2}(1 - \tanh^2(\epsilon u))$.

Soft Rounding(2) and HTGE (36) utilize the fact that:

$$R(x) = \lfloor x \rfloor + 0.5 + \frac{1}{2} S(x - (\lfloor x \rfloor + 0.5)) \qquad (29)$$

Rather than smoothing $R$, Soft Rounding and HTGE use tanh to smooth $S$ in Equation 29. As we have proved that tanh is a randomized smoothing variant of $S$, we omit further discussion of Soft Rounding and HTGE.

### D.4 Polynomial-based Functions

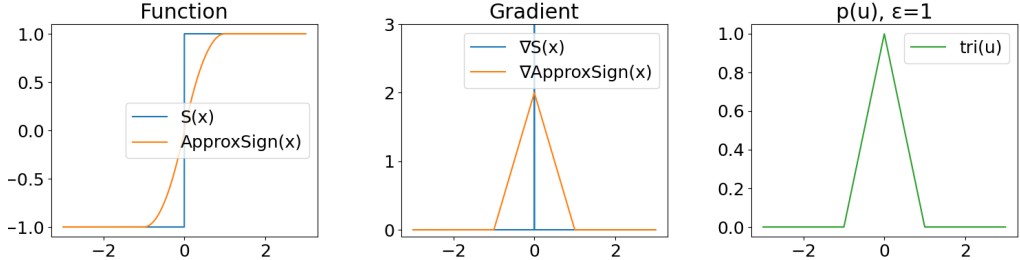

Figure 8: The ApproxSign function approximates the sign function.

The ApproxSign STE (26) uses the ApproxSign function, defined as:

$$\text{ApproxSign}(x) = \begin{cases} -1, & \text{if } x < -1 \\ 2x + x^2, & \text{if } -1 \leq x < 0 \\ 2x - x^2, & \text{if } 0 \leq x < 1 \\ 1, & \text{otherwise} \end{cases} \qquad (30)$$

Its derivative is defined as:

$$\nabla \text{ApproxSign}(x) = \begin{cases} 2 + 2x, & \text{if } -1 \le x < 0 \\ 2 - 2x, & \text{if } 0 \le x < 1 \\ 0, & \text{otherwise} \end{cases} \tag{31}$$

Statement to prove: $\text{ApproxSign}(x) = S_{\epsilon, p(u)}(x)$, where $p(u) = \epsilon \cdot \text{tri}(\epsilon u)$. The tri function is defined as:

$$\text{tri}(u) = \begin{cases} 1 - |u|, & \text{if } |u| \le 1 \\ 0, & \text{otherwise} \end{cases} \tag{32}$$

We know that

$$\begin{aligned} S_{\epsilon, \epsilon \cdot \text{tri}(\epsilon u)}(x) \\ &= E_{u \sim \epsilon \cdot \text{tri}(\epsilon u)}[S(x + \epsilon u)] \\ &= \int_{u=-1/\epsilon}^{1/\epsilon} S(x + \epsilon u) \epsilon \cdot \text{tri}(\epsilon u) du \end{aligned} \tag{33}$$

We substitute $z = \epsilon u$, $u = z/\epsilon$, and $du = dz/\epsilon$.

$$S_{\epsilon, \epsilon \cdot \text{tri}(\epsilon u)}(x) = \int_{z=-1}^{1} S(x + z)\text{tri}(z)dz \tag{34}$$

We now consider four cases:

- $x \ge 1$:
  The integration bound implies that $z \ge -1$; therefore, $x + z \ge 0$ and $S(z) = 1$.

  $$S_{\epsilon, \epsilon \cdot \text{tri}(\epsilon u)}(x) = \int_{-1}^{1} S(x + z)\text{tri}(z)dz = \int_{-1}^{1} \text{tri}(z)dz = 1 \tag{35}$$

- $x < -1$:
  The integration bound implies that $z \le 1$; therefore, $x + z < 0$ and $S(z) = -1$.

  $$S_{\epsilon, \epsilon \cdot \text{tri}(\epsilon u)}(x) = \int_{-1}^{1} S(x + z)\text{tri}(z)dz = \int_{-1}^{1} -\text{tri}(z)dz = -1 \tag{36}$$

- $-1 \le x < 0$:

  $$\begin{aligned} S_{\epsilon, \epsilon \cdot \text{tri}(\epsilon u)}(x) \\ &= \int_{-1}^{1} S(x + z)\text{tri}(z)dz \\ &= \int_{-1}^{0} S(x + z)\text{tri}(z)dz + \int_{0}^{-x} S(x + z)\text{tri}(z)dz + \int_{-x}^{1} S(x + z)\text{tri}(z)dz \\ &= \int_{-1}^{0} -\text{tri}(z)dz + \int_{0}^{-x} -\text{tri}(z)dz + \int_{-x}^{1} \text{tri}(z)dz \\ &= -0.5 + \int_{0}^{-x} -(1 - z)dz + \int_{-x}^{1} (1 - z)dz \\ &= -0.5 + \left[-z + \frac{1}{2}z^2\right]_{0}^{-x} + \left[z - \frac{1}{2}z^2\right]_{-x}^{1} \\ &= -0.5 + (x + \frac{1}{2}x^2) + 0.5 - (-x - \frac{1}{2}x^2) \\ &= 2x + x^2 \end{aligned} \tag{37}$$

- $0 \leq x < 1$:

$S_{\epsilon=1,\text{tri}(z)}(x)$

$$
\begin{aligned}
&= \int_{-1}^{1} S(x+z)\text{tri}(z)dz \\
&= \int_{-1}^{-x} S(x+z)\text{tri}(z)dz + \int_{-x}^{0} S(x+z)\text{tri}(z)dz + \int_{0}^{1} S(x+z)\text{tri}(z)dz \\
&= \int_{-1}^{-x} -\text{tri}(z)dz + \int_{-x}^{0} \text{tri}(z)dz + \int_{0}^{1} \text{tri}(z)dz \\
&= \int_{-1}^{-x} -(1+z)dz + \int_{-x}^{0} (1+z)dz + 0.5 \\
&= \left[ -z - \frac{1}{2}z^2 \right]_{-1}^{-x} + \left[ z + \frac{1}{2}z^2 \right]_{-x}^{0} + 0.5 \\
&= (x - \frac{1}{2}x^2) - 0.5 + -(-x + \frac{1}{2}x^2) + 0.5 \\
&= 2x - x^2 \quad \square
\end{aligned}
$$

(38)

To ensure $p(u)$ has unit variance, we can use $\epsilon = \frac{1}{\sqrt{6}}$ and $p(u) = \frac{1}{\sqrt{6}} \cdot \text{tri}(\frac{u}{\sqrt{6}})$, or equivalently the following:

$$
p(u) = \begin{cases} \frac{1}{\sqrt{6}}(1 - |\frac{u}{\sqrt{6}}|), & \text{if } |u| \leq \sqrt{6} \\ 0, & \text{otherwise} \end{cases}
$$

(39)

### D.5 Confidence-Guided Masking (CGM)

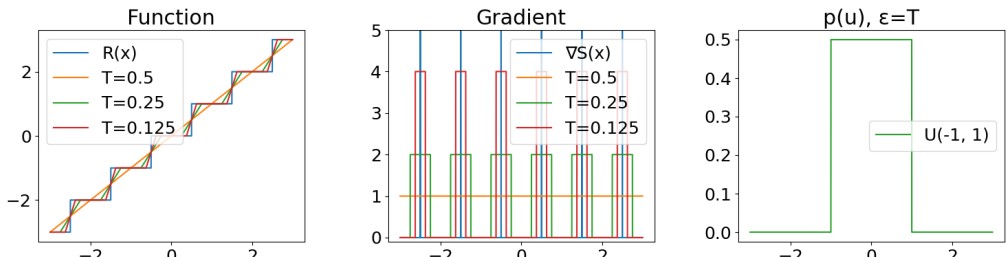

Figure 9: Confidence-Guided Masking approximates rounding.

Confidence-Guided Masking (25) uses a hyperparameter confidence threshold $0 < T \leq 0.5$, and is defined as follows:

$$
\nabla\text{CGM}(x) = \begin{cases} 0 & , \text{if } |x - R(x)| < 0.5 - T, \\ 1 & , \text{otherwise} \end{cases}
$$

(40)

This is equivalent to using the following gradient, but with the learning rate scaled by a factor of $2T$.

$$
\nabla\text{CGM}_{\text{scaled}}(x) = \begin{cases} 0 & , \text{if } |x - R(x)| < 0.5 - T, \\ \frac{1}{2T} & , \text{otherwise} \end{cases}
$$

(41)

We prove that $\text{CGM}_{\text{scaled}}$ is equivalent to a smoothed variant of $R$.

Statement to prove: $\nabla \text{CGM}_{\text{scaled}}(x) = \nabla R_{\epsilon=T,U(-1,1)}(x)$.

$$R_{\epsilon=T,U(-1,1)}(x)$$

$$= E_{u \sim U(-1,1)}[R(x+Tu)]$$

$$= \int_{-1}^{1} R(x+Tu)p(u)du$$

$$= \frac{1}{2}\int_{-1}^{1} R(x+Tu)du \tag{42}$$

$$= \frac{1}{2T}\int_{z=x-T}^{x+T} R(z)dz$$

- We first consider the case $x + T < \lfloor x \rfloor + 0.5$.
  This means $R(x-T) = R(x) = R(x+T) = \lfloor x \rfloor$ and $x - \lfloor x \rfloor = x - R(x) < 0.5 - T$, which is the positive half of $|x - R(x)| < 0.5 - T$.

$$R_{\epsilon=T,U(-1,1)}(x)$$

$$= \frac{1}{2T}\int_{z=x-T}^{x+T} R(z)dz$$

$$= \frac{1}{2T}\int_{z=x-T}^{x+T} \lfloor x \rfloor dz$$

$$= \frac{1}{2T}(\lfloor x \rfloor)((x+T)-(x-T)) \tag{43}$$

$$= \frac{1}{2T}(\lfloor x \rfloor)(2T)$$

$$= \lfloor x \rfloor$$

Thus, $\nabla R_{\epsilon=T,U(-1,1)}(x) = 0$ as the floor function is flat when $x + T < \lfloor x \rfloor + 0.5$.

- Next, we consider the case $x - T > \lfloor x \rfloor + 0.5$.
  This means $R(x-T) = R(x) = R(x+T) = \lceil x \rceil$ and $x - \lfloor x \rfloor > 0.5 + T$, or equivalently $x - R(x) > T - 0.5$, which is the negative half of $|x - R(x)| < 0.5 - T$.

$$R_{\epsilon=T,U(-1,1)}(x)$$

$$= \frac{1}{2T}\int_{z=x-T}^{x+T} R(z)dz$$

$$= \frac{1}{2T}\int_{z=x-T}^{x+T} \lceil x \rceil dz$$

$$= \frac{1}{2T}(\lceil x \rceil)((x+T)-(x-T)) \tag{44}$$

$$= \frac{1}{2T}(\lceil x \rceil)(2T)$$

$$= \lceil x \rceil$$

Thus $\nabla R_{\epsilon=T,U(-1,1)}(x) = 0$, as the ceiling function is flat when $x - T > \lfloor x \rfloor + 0.5$.

- Finally we consider the case $x - T \leq \lfloor x \rfloor + 0.5 \leq x + T$.
  This is exactly when $|x - R(x)| < 0.5 - T$ is *false*.

$$R_{\epsilon=T, U(-1,1)}(x)$$

$$= \frac{1}{2T} \int_{z=x-T}^{x+T} R(z) dz$$

$$= \frac{1}{2T} \int_{z=x-T}^{\lfloor x \rfloor + 0.5} \lfloor x \rfloor dz + \frac{1}{2T} \int_{z=\lfloor x \rfloor + 0.5}^{x+T} \lceil x \rceil dz$$

$$= \frac{1}{2T}(\lfloor x \rfloor)((\lfloor x \rfloor + 0.5) - (x - T)) + \frac{1}{2T}(\lceil x \rceil)((x + T) - (\lfloor x \rfloor + 0.5))$$

$$= \frac{1}{2T}(\lfloor x \rfloor)(\lfloor x \rfloor - x + 0.5 + T) + \frac{1}{2T}(\lceil x \rceil)(x - \lfloor x \rfloor - 0.5 + T))$$

$$= \frac{1}{2T}(\lfloor x \rfloor)(2T) + \frac{1}{2T}(x - \lfloor x \rfloor - 0.5 + T))$$

$$= \lfloor x \rfloor + \frac{1}{2T}(x - \lfloor x \rfloor - 0.5 + T))$$

$$\tag{45}$$

Thus $\nabla R_{\epsilon=T, U(-1,1)}(x) = \frac{1}{2T}$ $\square$

To ensure unit-variance perturbation, we can use $\epsilon = \frac{T}{\sqrt{3}}$ and $p(u) = U(-\sqrt{3}, \sqrt{3})$.

# E   Experiment Configurations

In this section, we discuss the configurations of our experiments.

## E.1   Shallow Networks: Comparing the STE, n-SPSA, and FOGZO

We use a batch size of 512 and a learning rate of 2e-3 * batch_size / 32. We train for 10 epochs on MNIST. We use the AdamW optimizer and the Cosine Annealing scheduler. We quantize only the weights and use a precision of 2 bits. For the quantization scale $\alpha$ shared among the 2 linear layers, we follow the initialization technique of LSQ (13) and set $\alpha_i = 2 \cdot \text{absmean}(\theta_i)/\sqrt{Q_P}$, where $\alpha_i$ is the quantization scale of layer $i$, and $\theta_i$ is the weights for layer $i$. We then perform a weighted average $\alpha = \frac{1}{d} \sum_{i=1}^{l} \alpha_i(\theta_i.\text{numel}())$ where $d = \sum_{i=1}^{l} \theta_i.\text{numel}()$, and $l$ is the number of linear layers. This allows us to obtain a quantization scale $\alpha$ that is acceptable to both linear layers. Before training, we compute $\alpha$ once and never modify it during training.

In terms of weight quantizer design, we mostly follow LSQ, except that $\alpha$ is fixed to reduce potential factors that can prevent us from clearly identifying the strengths and weaknesses of the STE, n-SPSA, and FOGZO.

$$\hat{\theta} = \alpha \cdot \text{round}(\text{clip}(\theta/\alpha, Q_N, Q_P)) \text{ where } Q_N = -2^{p-1} \text{ and } Q_P = 2^{p-1} - 1 \tag{46}$$

However, n-SPSA and FOGZO with large sample size $n$ are too expensive to calculate efficiently, even for a simple 2-layer MLP. As such, we use a parallelized implementation that performs $n$ forward passes in parallel. We implement this by duplicating the model weights $n$ times and perturbing each copy with a different sampled perturbation $u_i$ or $v_i$. During each forward pass, all $n$ copies are calculated in parallel to reduce computation time.

For the FLOPs estimation in Figure 1, we count the FLOPs of linear layers using the formula where, given an activation matrix with shape $(b, c)$, and weight matrix with shape $(c, o)$, the forward pass FLOPs is $2boc$ and the backward pass is $4boc$.

Without quantization, the 2-layer MLP can reach a training loss of 0.23. With 2-bit weight quantization and using the Straight-Through Estimator, the training loss increases to 2.02. Using FOGZO with $\beta = 0.999$ and $n = 1$, the training loss improves to 1.97.

## E.2   Deeper Networks: Comparing the STE and FOGZO

In this section, we discuss the configurations of experiments on deeper networks. All models with fewer than 30 million parameters are trained on Nvidia RTX 2080 Ti GPUs with 11 GB of memory.

All models with parameter counts between 30 million and 200 million are trained on Nvidia RTX 5090 with 32 GB of RAM. All models with more than 200 million parameters are trained on Nvidia A100 80GB. We follow the training recipes and hyperparameters of (18), (48), (53), (35). We make no changes to the training recipes with a few exceptions for simplicity: we do not use EMA evaluation; we do not use gradient accumulation and simply follow the linear scaling rule (16) to scale batch size and learning rate; we keep BatchNorm in training mode but freeze the running stats when we measure the loss of the perturbed model.

### E.2.1 FOGZO can Improve a Variety of STEs

We adapt the training recipe from (18) for Resnets using the source code from (37), the training recipe from (48) for DeiTs, and the training recipe from (53) for LLaMAs. We train Resnet models with a single RTX 2080 Ti, DeiT models with four RTX 2080 Tis, and LLaMA models with one RTX 2080 Ti. If RTX 2080 Ti cannot fit the batch size specified in the original training recipe, we iteratively reduce the batch size by 2x and learning rate by 2x until we no longer run out of memory. Similar to Section E.1, we use a non-learnable quantization scale $\alpha$ shared by all linear layers in a model, calculated with the weighted average of all LSQ quantization scale initialization. Since LSQ does not support 1-bit quantization, we instead adapt the quantization scale calculation from (28), which uses $\alpha_i = \text{absmean}(\theta_i)$, if the target STE (tanh and ApproxSign) is designed for 1-bit quantization. For Resnets, we additionally implement a zeroth-order-safe BatchNorm, which simply disables the update of running statistics during zeroth-order operations in Algorithm 1.

