# OpenReview forum: "Improving the Straight-Through Estimator with Zeroth-Order Information"
_NeurIPS.cc/2025/Conference — NeurIPS 2025 poster_

### Official Review · Reviewer_WuB8 · 2025-06-04

**Clarity:** 2
**Significance:** 2
**Originality:** 2
**Rating:** 4
**Confidence:** 2

**Summary:**

The paper addresses the challenges of training neural networks with quantized parameters by proposing First-Order-Guided Zeroth-Order Gradient Descent (FOGZO), a hybrid gradient estimator. FOGZO combines the Straight-Through Estimator (STE), which is computationally efficient but biased, with zeroth-order (ZO) methods, which are theoretically sound but computationally expensive. The authors theoretically analyze their approach, claiming reduced bias compared to STE and improved efficiency relative to ZO methods, supported by empirical evaluations across diverse neural network architectures.

**Questions:**

Please address the above-mentioned weaknesses.

**Ethical Concerns:**

["NO or VERY MINOR ethics concerns only"]

**Final Justification:**

The authors alleviated some of my concerns and I increased my score accordingly.

**Limitations:**

Yes

**Quality:**

2

**Strengths And Weaknesses:**

# Strengths

Integrating STE with ZO methods offers a relatively novel methodological perspective.

The paper provides detailed mathematical analyses demonstrating how FOGZO interpolates between biased STE gradients and unbiased ZO gradients. The paper evaluates multiple neural architectures on relevant datasets, providing some empirical evidence supporting their claims.

Improving quantization-aware training is relevant given the increasing demand for efficient neural networks in resource-constrained environments.

# Weaknesses

The paper is very hard to follow. The authors overuse fuzzy adverbs of frequency such as mostly, often, some, etc.

While experiments do indicate improvements over STE, the practical significance of some results is relatively limited (e.g. only 0.5 to 2% improvement on popular models like ResNet). Given that, despite being more efficient than pure zeroth-order methods, FOGZO still incurs a substantial computational increase (~50%) over STE, it's practical relevance might be limited. Moreover, the approach currently applies strictly to weight quantization, potentially reducing it's relevance in practice even further.

The experimental results seem to be limited to individual runs. Repreating each experiment multiple times and reporting mean and std would make results more robutst. Right now, the instances where FOGZO does improve over STE could be outliers, particularily in experiments where the difference to the baseline is not that large.

Quantization in Graph Neural Networks is not discussed, despite these architectures having gained importance over the recent years.

# Conclusion

This is a borderline paper to me. If the authors sufficiently address the weak empirical part, I might consider changing my score from borderline reject (3) to borderline accept (4), but not more given the hard-to-follow write-up and expected limited impact.

---

> ### Author Rebuttal · Authors · 2025-07-31
>
> We thank Reviewer WuB8 for reviewing our paper. We apologize for the poor writing quality. As authors, it is our job to clearly convey the message to readers, and we have not done a sufficiently good job. Since NeurIPS does not allow us to submit an updated version of the paper during the rebuttal, we promise to update the camera-ready to avoid overusing adverbs of frequency and make the paper easier to follow.
>
> ## Activation Quantization
> Although we initially planned to leave activation quantization for future work, all reviewers list activation quantization as a weakness/question. Upon further investigation, we present new evidence that FOGZO can be applied to SOTA Quantization-Aware Training methods with **both** weight and activation quantization. In addition to LSQ [14], we additionally test QuEST [36], a new SOTA QAT method published (in June 2025) after we submit FOGZO (in May 2025). QuEST introduces a hadamard transform before quantization, and can significantly outperform LSQ. We add FOGZO to the publicly available source code of QuEST following Algorithm (1). We compare QuEST against QuEST-FOGZO, and LSQ against LSQ-FOGZO, and report the evaluation perplexity. Note that the results below are training results of models with both weights and activations quantized to 2-bit, and we use a $\beta_{min}$ of 1 - 1e-10.
>
> We use the benchmarks in the QuEST source code: LLaMA-30m and LLaMA-50m. Note that LLaMA-30m has 95 million parameters and LLaMA-50m has 125 million parameters, which is 5x to 6x larger than the largest model we have used in the submitted paper. Following QuEST’s advice, we train llama-30m with 3 billion tokens, and llama-50m with 5 billion tokens.
>
> |Model|QuEST|QuEST-FOGZO|LSQ|LSQ-FOGZO|
> |:-----------:|:-----------:|:-----------:|:-----------:|:-----------:|
> |LLaMA-30m|37.75|**37.37**|39.06|**37.38**|
> |LLaMA-50m|32.28|**32.06**|33.28|**32.41**|
>
> Results show that FOGZO can improve both QuEST and LSQ. In addition, FOGZO can even shrink the perplexity difference between these two SOTA methods.
>
> ## Training Time
> Three reviewers have listed increased training time/computation overhead as a weakness of the method, and we agree that this is a valid concern. However, we note that there are certain use cases where model quality takes precedence over training time. For example, the operator of a large fleet of already deployed self-driving cars may find it more economical to invest extra computation time in training versus needing to upgrade hardware in existing cars if those are the only two options for improving accuracy/performance/safety.  Intellectually, we are interested in the limits of inference accuracy at low precision without regard to training cost and believe this direction has merits in such scenarios.
>
> While FOGZO introduces an additional training overhead of 2 forward passes per iteration, we show that the training time overhead can be reduced by using the STE at the beginning of training, and switch to FOGZO at the end of training. Concretely, this means we only execute lines 6-16 in Algorithm 1 if $t \ge r * T$, where $r$ is how much (70%, 80%, 90%, etc.) of the training is performed in STE mode. In the following table, we use LSQ to train llama-30m with 3 billion C4 tokens. We report evaluation perplexity and training time. While FOGZO alone incurs an overhead of 58% to tradeoff a perplexity decrease of 1.6, a mixing strategy that uses 90% of STE and 10% of FOGZO can still improve over the STE by a perplexity decrease of 0.4 with only a 6% increase in training time.
> | Method              | Evaluation Perplexity | Training Time (hours) |
> |---------------------|-----------------------|-----------------------|
> | STE                 | 39.06                 | 3.1                   |
> | 90% STE + 10% FOGZO | 38.67                 | 3.3                   |
> | 80% STE + 20% FOGZO | 38.38                 | 3.5                   |
> | 70% STE + 30% FOGZO | 37.93                 | 3.7                   |
> | FOGZO               | 37.38                 | 4.9                   |
>
> Below we extend the training time of the STE by adding more training data. We compare this pure-STE-with-more-data strategy and the previous mix-STE-FOGZO strategy under the same training time. Each row is a separate training run. Results show mix-STE-FOGZO outperforms. This also suggests FOGZO can have higher data efficiency than STE, as the FOGZO runs use less training data to achieve lower perplexity.
>
> | Method              | C4 Tokens (billion) | Evaluation Perplexity | Training Time (hours) |
> |---------------------|---------------------|-----------------------|-----------------------|
> | STE                 | 3.174               | 38.69                 | 3.3                   |
> | 90% STE + 10% FOGZO | 3                   | **38.67**                 | 3.3                   |
> | STE                 | 3.348               | 38.43                 | 3.5                   |
> | 80% STE + 20% FOGZO | 3                   | **38.38**                 | 3.5                   |
> | STE                 | 3.522               | 38.25                 | 3.7                   |
> | 70% STE + 30% FOGZO | 3                   | **37.93**                 | 3.7                   |
>
>
> ## Individual Runs of Experiment
>
> Below we report 4 rows of Table 2 with repeated runs and different random seeds. We are currently running experiments for the other rows in Table 2. Due to resource constraints, some of the larger experiments (such as the Imagenet-1k ones) require ~25 days. However, we will include all of Table 2 in the camera-ready.
>
> For each data point, we repeat the experiment 5 times and report mean and 2 standard deviations (covering about 96% probability mass). We report both training loss and evaluation accuracy. However, we note that training loss is the most direct indicator of gradient quality. Evaluation accuracy, on the other hand, is an indicator of both gradient quality and effectiveness of regularization techniques, which is not a focus of this work.
>
> | Experiment                                                | LSQ + Identity STE           | LSQ + Identity FOGZO         |
> |-----------------------------------------------------------|------------------------------|------------------------------|
> | DeiT-Tiny + Imagenet-100 (Training Loss / Eval Accuracy)  | 3.00±0.0166 / 72.76±0.31     | **2.98±0.012 / 72.92±0.60**  |
> | DeiT-Small + Imagenet-100 (Training Loss / Eval Accuracy) | 2.62±0.012 / 79.55±0.86      | **2.57±0.016 / 80.06±0.48**  |
> | Resnet-18 + Imagenet-100 (Training Loss / Eval Accuracy)  | 0.45±0.0084 / **80.23±0.84** | **0.44±0.0098** / 80.04±1.14 |
> | Resnet-50 + Imagenet-100 (Training Loss / Eval Accuracy)  | 0.43±0.0182 / 82.81±0.82     | **0.39±0.0086 / 83.67±0.26** |
> Lastly, we note that in the language model quantization community, reporting the standard deviation is not a convention, unlike in the vision model community. For example, papers [26,43,32,27,36] do not report standard deviation in their tables.  We hypothesize this is due to the prohibitive cost of running large models multiple times, and we are also experiencing the same issue.
>
> ## Questions
>
> 1. Quantization in Graph Neural Networks is not discussed, despite these architectures having gained importance over the recent years.
>
> Quantization methods for Graph Neural Networks typically utilize the fact that real-world graph dataset are sparse and nodes have different numbers of neighbours, and propose specialized quantization methods that reduce the quantization error based on the degree of nodes such as [a] and [b]. These methods can benefit from general-purpose quantization methods like LSQ and QuEST, which accelerates linear and convolutional layers regardless of the architecture of the model. FOGZO is designed to improve general-purpose quantization methods, by using their estimated gradient as a bias source to accelerate SPSA, while using SPSA to reduce the gradient bias of the STE.
>
> [a] Shyam A. Tailor, Javier Fernandez-Marques, & Nicholas D. Lane. DEGREE-QUANT: QUANTIZATION-AWARE TRAINING FOR GRAPH NEURAL NETWORKS
>
> [b] Zeyu Zhu, Fanrong Li, Zitao Mo, Qinghao Hu, Gang Li, Zejian Liu1,
> Xiaoyao Liang, &  Jian Cheng. A2Q: AGGREGATION-AWARE QUANTIZATION FOR GRAPH NEURAL NETWORKS

---

### Official Review · Reviewer_y8vC · 2025-06-28

**Clarity:** 3
**Significance:** 3
**Originality:** 3
**Rating:** 4
**Confidence:** 3

**Summary:**

This paper introduces ​​FOGZO​​, a novel method for QAT that synergistically combines the computational efficiency of the STE  with the unbiased gradients of ZO methods. FOGZO demonstrates significant empirical improvements over STE and ZO baselines across multiple architectures (ResNet, DeiT, LLaMA) and datasets (ImageNet, C4), with minimal computational overhead. The paper is well-written, technically sound, and contribute a new direction for optimizing STE.

**Questions:**

1. Could perturbing activations (not just weights) during ZO steps resolve the performance gap?
2. Or can your maintain the STE for activation scale training to see how FOGZO works with weight-activation quantization?
3. Could you compared your methods with EWGS? This is a paper using second-order information to suppress the STE, maybe both the efficiency and performance.
4. Does FOGZO’s gradient variance reduction amplify benefits from optimizers like AdamW or Lion?
5. Could you share the time cost and memory usage of your methods compared to STE?
6. I saw you added hardmard transform in your llama models, would this method be helpful with hardmard transform or can this improve the performance after transformed?
7. Could you share the FLOPs for different step-size of ZOs?

**Ethical Concerns:**

["NO or VERY MINOR ethics concerns only"]

**Final Justification:**

This paper presents a new approach to quantization gradient estimation. It looks novel to me as it uses zero-order optimization, which is the first time taking this approach into the gradient estimation for quantization. The experiments show the performance improvement of this method. However, the increase in training time is too significant to make it feasible for a real-world situation. It's still a toy.

Combining the rebuttal of the authors, I would give a borderline accept.

**Limitations:**

Yes

**Quality:**

3

**Strengths And Weaknesses:**

Strengths​:
- The combination of STE and ZO gradients​​ via a biased-unbiased gradient mixing strategy is novel and new for me.
- The theoretical analysis is detailed, showing the effective of FOGZO.
- Comprehensive experiments​​ on vision and language models show consistent gains of FOGZO.
- The paper is well-written and easy to follow.

Weaknesses:
- The searching for hyperparameter beta is time-cost.
- Although FOGZO reduces the time cost compared to ZO based method, but it is still time cost, narrowing the use range for large models.
- It doesn't work good with weight-activation quantization.

---

> ### Author Rebuttal · Authors · 2025-07-31
>
> We thank Reviewer y8vc for reviewing our paper.
>
> ## Activation Quantization
> Although we initially planned to leave activation quantization for future work, all reviewers list activation quantization as a weakness/question. Upon further investigation, we present new evidence that FOGZO can be applied to SOTA Quantization-Aware Training methods with **both** weight and activation quantization. In addition to LSQ [14], we additionally test QuEST [36], a new SOTA QAT method published (in June 2025) after we submit FOGZO (in May 2025). QuEST introduces a hadamard transform before quantization, and can significantly outperform LSQ. We add FOGZO to the publicly available source code of QuEST following Algorithm (1). We compare QuEST against QuEST-FOGZO, and LSQ against LSQ-FOGZO, and report the evaluation perplexity. Note that the results below are training results of models with both weights and activations quantized to 2-bit, and we use a $\beta_{min}$ of 1 - 1e-10.
>
> We use the benchmarks in the QuEST source code: LLaMA-30m and LLaMA-50m. Note that LLaMA-30m has 95 million parameters and LLaMA-50m has 125 million parameters, which is 5x to 6x larger than the largest model we have used in the submitted paper. Following QuEST’s advice, we train llama-30m with 3 billion tokens, and llama-50m with 5 billion tokens.
>
> |Model|QuEST|QuEST-FOGZO|LSQ|LSQ-FOGZO|
> |:-----------:|:-----------:|:-----------:|:-----------:|:-----------:|
> |LLaMA-30m|37.75|**37.37**|39.06|**37.38**|
> |LLaMA-50m|32.28|**32.06**|33.28|**32.41**|
>
> Results show that FOGZO can improve both QuEST and LSQ. In addition, FOGZO can even shrink the perplexity difference between these two SOTA methods.
>
> ## Training Time
> Three reviewers have listed increased training time/computation overhead as a weakness of the method, and we agree that this is a valid concern. However, we note that there are certain use cases where model quality takes precedence over training time. For example, the operator of a large fleet of already deployed self-driving cars may find it more economical to invest extra computation time in training versus needing to upgrade hardware in existing cars if those are the only two options for improving accuracy/performance/safety.  Intellectually, we are interested in the limits of inference accuracy at low precision without regard to training cost and believe this direction has merits in such scenarios.
>
> While FOGZO introduces an additional training overhead of 2 forward passes per iteration, we show that the training time overhead can be reduced by using the STE at the beginning of training, and switch to FOGZO at the end of training. Concretely, this means we only execute lines 6-16 in Algorithm 1 if $t \ge r * T$, where $r$ is how much (70%, 80%, 90%, etc.) of the training is performed in STE mode. In the following table, we use LSQ to train llama-30m with 3 billion C4 tokens. We report evaluation perplexity and training time. While FOGZO alone incurs an overhead of 58% to tradeoff a perplexity decrease of 1.6, a mixing strategy that uses 90% of STE and 10% of FOGZO can still improve over the STE by a perplexity decrease of 0.4 with only a 6% increase in training time.
> | Method              | Evaluation Perplexity | Training Time (hours) |
> |---------------------|-----------------------|-----------------------|
> | STE                 | 39.06                 | 3.1                   |
> | 90% STE + 10% FOGZO | 38.67                 | 3.3                   |
> | 80% STE + 20% FOGZO | 38.38                 | 3.5                   |
> | 70% STE + 30% FOGZO | 37.93                 | 3.7                   |
> | FOGZO               | 37.38                 | 4.9                   |
>
> Below we extend the training time of the STE by adding more training data. We compare this pure-STE-with-more-data strategy and the previous mix-STE-FOGZO strategy under the same training time. Each row is a separate training run. Results show mix-STE-FOGZO outperforms. This also suggests FOGZO can have higher data efficiency than STE, as the FOGZO runs use less training data to achieve lower perplexity.
>
> | Method              | C4 Tokens (billion) | Evaluation Perplexity | Training Time (hours) |
> |---------------------|---------------------|-----------------------|-----------------------|
> | STE                 | 3.174               | 38.69                 | 3.3                   |
> | 90% STE + 10% FOGZO | 3                   | **38.67**                 | 3.3                   |
> | STE                 | 3.348               | 38.43                 | 3.5                   |
> | 80% STE + 20% FOGZO | 3                   | **38.38**                 | 3.5                   |
> | STE                 | 3.522               | 38.25                 | 3.7                   |
> | 70% STE + 30% FOGZO | 3                   | **37.93**                 | 3.7                   |
>
>
> ## Questions
> 1. Could perturbing activations (not just weights) during ZO steps resolve the performance gap?
>
> Our initial thought is to use activation perturbation. However, this requires capturing the STE-estimated gradient of activations, which can be prohibitively expensive to store. Upon further investigation, we found that FOGZO can be used as is for activation quantization as noted above.
>
> 2. Or can your maintain the STE for activation scale training to see how FOGZO works with weight-activation quantization?
>
> Yes, this is exactly how we produce our new results above. We use the STE for both activation and weight quantizers, and simply apply FOGZO as is following Algorithm 1.
>
> 3. Could you compared your methods with EWGS? This is a paper using second-order information to suppress the STE, maybe both the efficiency and performance.
>
> We find EWGS to be an elegant method that addresses the flaws of the STE in an orthogonal direction. We believe EWGS and FOGZO may benefit from each other. This is because FOGZO assumes nothing about its source of bias/the chosen STE, only that it is a “mostly accurate” gradient estimator. In fact, the computation reduction effect of FOGZO relies on this prior belief that the source of bias is sufficiently accurate. Without a proper high-quality source of bias, FOGZO reduces to SPSA, which suffers from large gradient variance when computation is limited. Since EWGS uses second-order information to produce better quality descent directions, EWGS may be used as a better-than-STE source of bias for FOGZO. In addition, we note that as in Figure 1, FOGZO is by no means a perfect gradient estimator compared to SPSA with unlimited computation. Therefore, there is still a large accuracy gap that can be obtained with better sources of bias like EWGS.
>
> Unfortunately, we are unable to complete experiments on EWGS by the rebuttal deadline due to resource limitations. We will be sure to at least cite and discuss EWGS in the camera-ready version.
>
> 4. Does FOGZO’s gradient variance reduction amplify benefits from optimizers like AdamW or Lion?
>
> We note that FOGZO’s gradient variance reduction is compared to SPSA. Compared to the STE, we find that FOGZO still generates slightly larger gradient variance due to $\beta_{min}$ not being exactly 1, as shown in Figure 1 empirically and discussed theoretically in Section 3.2. Therefore, we believe FOGZO actually benefits from the gradient variance reduction effect of optimizers like Adam. We use AdamW by default for experiments that use models with the Transformer architecture (such as DEIT and LLaMA). We also used AdamW for the experiments in Figure 1.
>
> 5. Could you share the time cost and memory usage of your methods compared to STE?
>
> Below we report the training time on the largest model, one from each model architecture. As noted above, we show that FOGZO’s overhead can be reduced with a mixing strategy between STE and FOGZO.
>
> | Model      | Data                   | GPU         | STE (hour) | FOGZO (hour) |
> |------------|------------------------|-------------|------------|--------------|
> | Resnet-50  | 90 epoch Imagenet-100  | RTX 2080 Ti | 13.6       | 24.6         |
> | DeIT-Small | 300 epoch Imagenet-100 | RTX 2080 Ti | 14.7       | 23.6         |
> | LLaMA-50m  | 5B C4 token            | RTX 5090    | 6.5        | 10.2         |
>
> Below we report the training memory on the largest model, one from each model architecture. We do not observe any difference in training memory. We believe this is because the bottleneck of training memory is the backward pass. Once the backward pass is completed, most of the intermediate results are freed, and FOGZO’s subsequent two forward passes has the same training memory demand of inference.
>
> |            | STE (GB) | FOGZO (GB) |
> |------------|----------|------------|
> | Resnet-50  | 7.3      | 7.3        |
> | DeIT-Small | 10.2     | 10.2       |
> | LLaMA-50m  | 19.9     | 19.9       |
>
> 6. I saw you added hardmard transform in your llama models, would this method be helpful with hardmard transform or can this improve the performance after transformed?
>
> Yes, FOGZO can improve the performance when hadamard transform is used as noted in the results above with QuEST.
>
> 7. Could you share the FLOPs for different step-size of ZOs?
>
> We do not quite understand this question. By step size, are you referring to the learning rate? Is this the FLOPs measurement in Figure 1?

---

> > ### Comment · Reviewer_y8vC · 2025-08-03
> > **Thanks.**
> >
> > Thanks for solving my concerns.
> >
> > For the Q7, the Figure 1 FLOPs measurement is enough.
> >
> > It's an interesting try at taking ZO as a gradient estimator. Including the training time in your next version paper would help the reader better understand your method.

---

> > > ### Author Response · Authors · 2025-08-03
> > > **Thank You**
> > >
> > > We again thank the reviewer for reviewing our paper and providing valuable feedback. We will be sure to include training time in the next version of our paper.

---

### Official Review · Reviewer_j9sv · 2025-07-01

**Clarity:** 4
**Significance:** 3
**Originality:** 4
**Rating:** 4
**Confidence:** 5

**Summary:**

The paper proposes FOGZO, a novel method that combines the Straight-Through Estimator (STE) and Zeroth-Order (ZO) gradient descent for training quantized neural networks. While STE is efficient but biased, and ZO methods are unbiased but computationally expensive, FOGZO leverages STE as a bias source to guide low-cost ZO updates, suppressing STE’s flaws. The method introduces a mixing parameter to balance biased and unbiased gradients. Experiments show FOGZO improves training accuracy and efficiency across various models, including ResNet, DeiT, and LLaMA, outperforming both STE and ZO baselines.

**Questions:**

1. Figure 1 is hard to understand. Why are reciprocals of loss and flops used? They make readers hard to clearly and easily figure out which line is better.
2. The chosen $\beta$ value is very close to 1, implying that the STE part dominates the ZO part. It is better to call it ZO Guided STE? In addition, as $\beta$ is very close to 1 and $1-\beta$ is very close to 0. Does it cause any numerical issues during training, especially in FP16/BF16 training?

**Ethical Concerns:**

["NO or VERY MINOR ethics concerns only"]

**Quality:**

3

**Strengths And Weaknesses:**

Strengths:
1. Introduces FOGZO, a first-order-guided zeroth-order gradient estimator that leverages the biased yet efficient STE to guide the unbiased zeroth-order optimization.
2. Provides theoretical analysis showing that FOGZO can suppress incorrect STE gradients while preserving computational efficiency.
3. Performed numerical experiments to demonstrate the theoretical findings.

Weakness:
1. Increased the training costs and training time.
2. More hyperparameters are introduced, which may require more careful tuning.
3. Experiments are primarily conducted on small-scale models. Gains with SOTA QAT methods, like LSQ, are marginal for most models except DeiT-Tiny.

---

> ### Author Rebuttal · Authors · 2025-07-31
>
> We thank Reviewer j9sv for reviewing our paper.
>
> ## Small Scale Models and Marginal Gains with SOTA QAT Methods
>
> We provide new evidence that FOGZO can work well with larger models: LLaMA-30m and LLaMA-50m. Note that LLaMA-30m has 95 million parameters and LLaMA-50m has 125 million parameters, which is 5x to 6x larger than the largest model (~20 million) we have used in the submitted paper.
>
> In addition to LSQ [14], we additionally test QuEST [36], a new SOTA QAT method published (in June 2025) after we submit FOGZO (in May 2025). QuEST introduces a Hadamard transform before quantization, and can significantly outperform LSQ.
> We add FOGZO to the publicly available source code of QuEST following Algorithm (1).  Following QuEST’s advice, we train llama-30m with 3 billion tokens, and llama-50m with 5 billion tokens. We compare QuEST against QuEST-FOGZO, and LSQ against LSQ-FOGZO, and report the evaluation perplexity. Note that the results below are training results of models with both weights and activations quantized to 2-bit, and we use a $\beta_{min}$ of 1 - 1e-10.
>
> |Model|QuEST|QuEST-FOGZO|LSQ|LSQ-FOGZO|
> |:-----------:|:-----------:|:-----------:|:-----------:|:-----------:|
> |LLaMA-30m|37.75|**37.37**|39.06|**37.38**|
> |LLaMA-50m|32.28|**32.06**|33.28|**32.41**|
>
> Results show that FOGZO can improve both QuEST and LSQ. In addition, FOGZO can even shrink the perplexity difference between these two SOTA methods.
>
> ## Training Time
> Three reviewers have listed increased training time/computation overhead as a weakness of the method, and we agree that this is a valid concern. However, we note that there are certain use cases where model quality takes precedence over training time. For example, the operator of a large fleet of already deployed self-driving cars may find it more economical to invest extra computation time in training versus needing to upgrade hardware in existing cars if those are the only two options for improving accuracy/performance/safety.  Intellectually, we are interested in the limits of inference accuracy at low precision without regard to training cost and believe this direction has merits in such scenarios.
>
> While FOGZO introduces an additional training overhead of 2 forward passes per iteration, we show that the training time overhead can be reduced by using the STE at the beginning of training, and switch to FOGZO at the end of training. Concretely, this means we only execute lines 6-16 in Algorithm 1 if $t \ge r * T$, where $r$ is how much (70%, 80%, 90%, etc.) of the training is performed in STE mode. In the following table, we use LSQ to train llama-30m with 3 billion C4 tokens. We report evaluation perplexity and training time. While FOGZO alone incurs an overhead of 58% to tradeoff a perplexity decrease of 1.6, a mixing strategy that uses 90% of STE and 10% of FOGZO can still improve over the STE by a perplexity decrease of 0.4 with only a 6% increase in training time.
> | Method              | Evaluation Perplexity | Training Time (hours) |
> |---------------------|-----------------------|-----------------------|
> | STE                 | 39.06                 | 3.1                   |
> | 90% STE + 10% FOGZO | 38.67                 | 3.3                   |
> | 80% STE + 20% FOGZO | 38.38                 | 3.5                   |
> | 70% STE + 30% FOGZO | 37.93                 | 3.7                   |
> | FOGZO               | 37.38                 | 4.9                   |
>
> Below we extend the training time of the STE by adding more training data. We compare this pure-STE-with-more-data strategy and the previous mix-STE-FOGZO strategy under the same training time. Each row is a separate training run. Results show mix-STE-FOGZO outperforms. This also suggests FOGZO can have higher data efficiency than STE, as the FOGZO runs use less training data to achieve lower perplexity.
>
> | Method              | C4 Tokens (billion) | Evaluation Perplexity | Training Time (hours) |
> |---------------------|---------------------|-----------------------|-----------------------|
> | STE                 | 3.174               | 38.69                 | 3.3                   |
> | 90% STE + 10% FOGZO | 3                   | **38.67**                 | 3.3                   |
> | STE                 | 3.348               | 38.43                 | 3.5                   |
> | 80% STE + 20% FOGZO | 3                   | **38.38**                 | 3.5                   |
> | STE                 | 3.522               | 38.25                 | 3.7                   |
> | 70% STE + 30% FOGZO | 3                   | **37.93**                 | 3.7                   |
>
>
> ## Using Reciprocals in Figure 1
> Two reviewers noted using the reciprocal of training loss and flops makes the figure hard to read. We sincerely apologize for the inconvenience. We thought reviewers might prefer a graph where the top-right corner is the best position. We now realized this was clearly a bad choice and we will remove the reciprocals in the camera-ready, making the lower-left corner the best position.
>
> ## Questions
>
> 1. The chosen β value is very close to 1, implying that the STE part dominates the ZO part. It is better to call it ZO Guided STE? In addition, as β is very close to 1 and 1−β is very close to 0. Does it cause any numerical issues during training, especially in FP16/BF16 training?
>
> For clarification, training with the STE uses gradient g, and training with FOGZO and a β value of exactly 1 uses a gradient that is approximately $\hat{g} \hat{g}^T \nabla L_{smooth}$ (as shown in Equation 8). This means even when β is exactly one, FOGZO can still suppress the STE gradient if it is inaccurate, which may or may not be a good thing. If the STE is only occasionally inaccurate, FOGZO can prevent the STE from making mistakes when it is wrong, resulting in improved model quality. However, if the STE is constantly inaccurate, FOGZO will constantly suppress the STE gradient, and training can stop making progress. As shown in Figure 2, when FOGZO is used with a β value of exactly 1, outperforming the STE is not guaranteed (but it can happen). Therefore, we believe that although we use a β value close to 1, FOGZO is still fundamentally a zeroth-order method (as it has an incorrect gradient suppression effect similar to SPSA). However, we agree that “ZO Guided STE” and “STE/FO guided ZO” are very similar to one another, and both are accurate descriptions to some degree.
>
> Most of our results are trained in FP32 precision. However, in the QuEST/LSQ/training time experiments above, we use torch.amp.autocast to perform bfloat16 training. We did not observe any numerical issues.

---

> > ### Comment · Reviewer_j9sv · 2025-08-03
> >
> > Thanks a lot for responding to my concerns!
> >
> > I still have concerns about the scalability of the proposed method, as only very small-scale models are tested in the paper. Larger models are much more widely used in practice and also have much higher demands for compression. In addition, it is known that some algorithms may have very different performance for larger models and/or more training steps. It is important to have some tests on larger models, like 3B or 7B models, to demonstrate the scalability.
> >
> > Another concern comes from the $\beta$ value. The chosen $1-\beta$ value is too tiny in the experiments, which is very uncommon in my understanding. As a result, the unbiased term is very tiny, which may even be smaller than round-off errors when using BF16 for training. It would be better to explain more about why such a tiny correction term can work and whether the $(1-\beta)\nabla \hat{L}_{smooth}(\theta)$ is correctly calculated.
> >
> > Given some concerns still persist, I will keep my original score. Thank you again for your responses.

---

> > > ### Author Response · Authors · 2025-08-04
> > >
> > > Thank you for the followup.
> > > - It is important to have some tests on larger models, like 3B or 7B models, to demonstrate the scalability.
> > >     - We acknowledge that testing whether a technique scales with model size is important.  Quantization-Aware Training (QAT) methods typically takes the same training time as training a full-precision model from scratch (e.g. LSQ [14] trains Resnet on Imagenet with 90 epochs, and QuEST [36] trains LLaMA following the 100-token-per-parameter scaling law). Training models (from scratch) at the 3B or 7B scale can be done, but is expensive.  For example, our estimate is that training a 3B model may take at least 3 months on a single high-end GPU like RTX 4090.  While we do have access to multi-GPU hardware, shared with others at our institution, we estimate that we would not be able to complete a 3B-scale model by the author-reviewer discussion deadline (we believe it would require on the order of multiple weeks without any interruption on the best hardware available to us).
> > >     - While some existing papers [26,43,32,27] report results for larger models, we note that these papers use Post-Training Quantization (PTQ) techniques and do not train from scratch. FOGZO, on the other hand, is a QAT technique optimized for training deep neural networks from scratch under the constraint of quantization noise.  QAT methods can produce low-precision models that have higher-quality than PTQ methods, but one of the caveats is they require training.
> > > - It would be better to explain more about why such a tiny correction term can work and whether the (1−β)∇L^smooth(θ) is correctly calculated.
> > >     - We note that the QuEST and LSQ experiments in the rebuttal are conducted using torch.amp.autocast, which performs all expensive operations (e.g. matrix multiplication/convolution) in the forward and backward pass using BF16 precision, and the inexpensive/error-prone operations (e.g. layernorm) in FP32. This type of mixed-precision training can achieve throughput identical to bf16 training. However, when using torch.amp.autocast, the weights themselves are still kept in FP32.
> > >     - We note that mixed-precision training methods are a standard approach to train (not finetune with LoRA) large DNNs. For example, paper “DeepSeek-v3 Technical Report” (Feb 2025) shows, in Figure 6 on Page 15, that even though the forward and backward pass are computed in BF16/FP8, the master weights are still stored in FP32. This is likely to prevent loss of information when updating the weights with small gradients.
> > >     - In our QuEST and LSQ experiments in the rebuttal, weights are kept in FP32 following a standard mixed-precision training approach (expensive computations are done in BF16). Therefore, we do not observe any numerical issues when FOGZO perturbs the weight as shown in Algorithm 1 Lines 10, 12, and 14.
> > >     - Next, we discuss the challenges of using FOGZO when weights are kept in BF16. We note that (1−β)∇L^smooth(θ) from Equation (8) is a mathematical heuristic derivation that provides us insight on the behaviour of FOGZO. How FOGZO actually works in practice, is by perturbing the weights with a quantity of $\epsilon v_i$. In practice, $\epsilon v_i$ can be calculated in FP32, and therefore regardless of the value of $\beta$ (whether it is close to 1), no information loss occurs at this stage. The challenge lies in the subsequent step when $\epsilon v_i$ is added to BF16 weights.
> > >     - In the following table we present results for both LSQ and LSQ-FOGZO, on Llama-30m, with BF16 weights and FP32 weights.
> > > | Computation | Weights | LSQ   | LSQ-FOGZO |
> > > |-------------|---------|-------|-----------|
> > > | BF16        | FP32    | 39.06 | **37.38**     |
> > > | BF16        | BF16    | 49.40 |  **48.84**  |
> > >    - As shown, both LSQ and LSQ-FOGZO suffer from numerical issues when using BF16 weights compared to using FP32 weights. While LSQ-FOGZO can **still** outperform LSQ, the perplexity gap has shrunk significantly when weights are BF16, since LSQ-FOGZO perturbs/updates weights more often than LSQ. This is due to the challenge of modifying BF16 weights with small updates. Because of this challenge, keeping weights in BF16 is not yet a mature technology as it has not been incorporated into SOTA LLM pre-training pipelines (e.g. DeepSeek-v3). As the field evolves, new techniques will be developed to reduce the error when updating BF16 weights (e.g. stochastic rounding as per the paper “Revisiting BFloat16 Training. Zamirai et. al”). These techniques can then be used to relieve the numerical issues of FOGZO.
> > >    - We will discuss the impact of BF16 weights in our Limitations section. However, again we note that as the field evolves, this limitation will eventually be addressed. Lastly, as discussed in the rebuttal, the goal of FOGZO is to advance the limits of inference accuracy at low precision without regard to training cost. We believe FOGZO has merits in scenarios where inference quality/efficiency outweighs training cost.

---

> > > > ### Comment · Reviewer_j9sv · 2025-08-04
> > > >
> > > > Thank you so much for your prompt responses!
> > > >
> > > > 1. I agree that the QAT method is much more expensive than the PTQ methods since QAT requires training. I was not asking for the QAT method to pre-train a model from scratch, which is certainly not realistic. Rather than training the entire model from scratch, a common practice is to apply the method to the pre-trained model on a calibration dataset, like the Wikipedia dataset, for the quantization purpose. For domain-specific tasks, the domain-specific small training dataset can be used. This usually can be done in a single GPU within a few hours or at most a few days.
> > > >
> > > > The latest QAT methods have conducted experiments in this way. You may check [1] [2] for more details. It would also be great to add a comparison with these SOTA QAT methods for LLMs.
> > > >
> > > > [1] Choi, Euntae, Sumin Song, Woosang Lim, and Sungjoo Yoo. "Rotate, Clip, and Partition: Towards W2A4KV4 Quantization by Integrating Rotation and Learnable Non-uniform Quantizer." arXiv preprint arXiv:2502.15779 (2025).
> > > > [2] Du, Dayou, Yijia Zhang, Shijie Cao, Jiaqi Guo, Ting Cao, Xiaowen Chu, and Ningyi Xu. "Bitdistiller: Unleashing the potential of sub-4-bit llms via self-distillation." arXiv preprint arXiv:2402.10631 (2024).
> > > >
> > > > 2. Thank you for the explanation. I was wondering why such a small perturbation can improve the model performance. When $1-\beta$ is very close to 0, the term $(1-\beta)\hat{\nabla}_{smooth}L(\theta)$ is nearly a noise term whose magnitude can be much less than the first term in (8). Why does such a small term improve the convergence?

---

> > > > > ### Author Response · Authors · 2025-08-06
> > > > >
> > > > > We will refer to papers [1] and [2] in your message as [a] and [b] to differentiate from the other references already included in the submitted paper.
> > > > > - Rather than training the entire model from scratch, a common practice is to apply the method to the pre-trained model
> > > > >     - Thank you for your clarification and for providing references to [a] and [b] which we were unaware of previously and have now read.  To clarify, we agree that fine-tuning of pre-trained models at the scale of 3B and 7B is achievable in the order of hours or days and indeed this at least partly explains the recent interest in PTQ.
> > > > >     - In responding to your suggestion we believe it may be helpful to taxonomize QAT into techniques that focus on pretraining versus fine-tuning.  Below, we will refer to the former as Quantization-Aware pre-Training (QApT) [36] and the latter as Quantization-Aware fine-Tuning (QAfT) [a,b]. QApT typically initializes a model from scratch and trains for iterations identical to full-precision pre-training [36]. Similar to [a][b], QAfT initializes from a pre-trained full-precision checkpoint, typically trains for much shorter durations/iterations, and may be used after first applying PTQ methods. QApT can have higher training efficiency [25] compared to first pre-training in full precision and subsequently perform QAfT. This is because when both activation and weights are quantized, the forward passes in QApT may be performed in low precision [25], leading to higher pre-training throughput and less pre-training time [50].  In addition, the training recipes (e.g. learning rate) and dynamics are different between pre-training and fine-tuning, and methods suitable for one may not be suitable for the other (e.g. LoRA).
> > > > >     - We note that the use of ZO methods for fine-tuning was studied in MeZO [34] and QuZO [54]. MeZO showed both theoretically and empirically that SPSA with low computation suffers less from gradient variance during fine-tuning (compared to pre-training), as when a model is well-trained, its Hessian is low-rank. Based on this insight, MeZO demonstrates fine-tuning performance comparable to (although not as good as) FO. One of FOGZO’s innovations is to bring ZO into the pre-training regime. The challenge we were dealing with when we designed FOGZO, is that SPSA with low computation suffers from large gradient variance in pre-training. As such, FOGZO introduces first-order guidance to reduce the gradient variance of SPSA. Therefore, FOGZO is fundamentally designed to solve the challenges of QApT. For QAfT, there are much better alternatives like PV-Tuning [33], which does not require FOGZO’s additional overhead of two forward passes per iteration and can still outperform the STE.
> > > > >     - We next discuss a challenge when applying FOGZO to QAfT. First, while BF16 weights are arguably not yet mature in pre-training as BF16 weights have not been incorporated into SOTA LLM pre-training pipelines (see our previous message from Aug 3), it is a common practice to perform fine-tuning with BF16 weights (as in [b]). However, as demonstrated in our previous response from August 3rd, FOGZO suffers more from using BF16 weights since it updates/perturbs weights more often.
> > > > >     - Nevertheless, we conducted experiments using the source code of BitDistiller [b] (we could not find the source code for [a]). We fine-tune a LLaMA-3B (the largest we can fit on our GPUs with BF16 weights) with 3 billion parameters on a teacher-generated dataset (as discussed in [b]) with 6400 sentences. For comparison, the much smaller QuEST/LSQ version of LLaMA-30m with 95 million parameters was initialized from scratch and pre-trained on 3 billion tokens (much more data). We report the evaluation loss in terms of Confidence-Aware Kullback-Leibler Divergence (CAKLD), a loss function proposed in [b].
> > > > > | BitDistiller | BitDistiller-FOGZO-bmin=1 |
> > > > > |--------------|--------------------|
> > > > > | **186.08**       | 195.01  |
> > > > > FOGZO with a $\beta_{min}$ of exactly 1 typically matches (is close to) the performance of the STE in our previous experiment with FP32 weights, since at this beta value FOGZO completely relies on the STE. However, when combined with BitDistiller, even a $\beta_{min}$ of exactly 1 underperforms due to numerical issues. We observe that the rounding error introduced in a perturbation/addition operation $\theta + \epsilon v$ has an average of 90% of the magnitude of the individual elements of the perturbation themselves ($\epsilon v$).
> > > > >     - We will add clarification in the introduction section that the target research problem is Quantization-Aware pre-training (or QAT over a large dataset) and explain that QA fine-tuning is a different problem (both fields tend to use the same terminology “QAT”). We will also explain in the Limitation section that FOGZO is not suitable for QA fine-tuning, and alternatives like PV-Tuning and QuZO should be used instead.
> > > > > - Why does such a small term improve the convergence?
> > > > >     - to be continued.

---

> > > > > > ### Author Response · Authors · 2025-08-06
> > > > > >
> > > > > > - Why does such a small term improve the convergence?
> > > > > >     - continued from the last message
> > > > > >     - While the term (1−β)∇L^smooth(θ) is small, it affects all weights in the model. Therefore, the cumulative effect on training dynamics is significant as the model grows in size. Larger networks are known to be more sensitive to weight updates as they are usually paired with smaller learning rates [36].  In addition, the term (1−β)∇L^smooth(θ) is applied every iteration and gradually increased in magnitude. Therefore, the overall impact of the term grows over long durations of pre-training. In our experiments, we use a (1−β) of 1e-2 for our 2-layer MLP with 0.008M weights, 1e-9 for DeiT-T with 5M weights, and 1e-10 for LLaMA-30m with 95M weights. While it seems (1-β) is shrinking, we believe the cumulative effect on training dynamics stays at the same order, as model size is also growing. As a future work, it might be possible to experimentally obtain a heuristic method to set (1-β) based on the number of parameters.

---

> > > > > > > ### Comment · Reviewer_j9sv · 2025-08-08
> > > > > > >
> > > > > > > Thank you so much for your responses!
> > > > > > >
> > > > > > > 1. I see. That makes sense if the algorithm is specifically developed for QAT pre-training. It would be helpful to clarify this in the paper. From the same perspective, including some larger-scale experiments would substantially strengthen the work. I understand that pre-training runs are costly, but as models scale, it is often observed that certain methods behave quite differently at various training stages and across different model sizes.
> > > > > > >
> > > > > > > 2. Thank you for the clarification! It is interesting to see this phenomenon. It seems like the beta value decreases very rapidly as the size increases.  Based on the observation, it would be better to see how the algorithm performs and how the beta value changes when the model scales up and the training becomes longer.

---

### Official Review · Reviewer_tcg3 · 2025-07-02

**Clarity:** 3
**Significance:** 2
**Originality:** 3
**Rating:** 4
**Confidence:** 4

**Summary:**

The straight through estimator (STE) is a commonly used work-around to circumvent the non-differentiability of rounding operators, and is frequently used in quantization aware training of neural networks. This paper proposes blending the STE with zero-order gradient estimation, essentially by taking their weighted sum as the gradient estimate. In particular, the method relies on perturbing the current parameter estimate in a random direction in order to obtain the zero-th order estimate. The proposed method is given a heuristic derivation,  as are the probability distributions associated with different straight through estimators,  and the method is numerically tested by training neural networks of various sizes and architectures.

**Questions:**

- Section 2.2: $u$ is a vector, so when you say it is unit variance, do you mean it is isotropic, i.e., having an identity as its covariance matrix?
- Eq (4) looks a little different from the definitions used in the n-SPSA papers [45, 46]. Is the difference cosmetic or substantial?
- "While some believe the STE is completely unjustifiable" -- who? what are there arguments?
- Eq (5) the multiplication by $s_i$ seems to suggest that you have a 50/50 chance of biasing your update in the negative direction to that of $\hat{g}$. Is there an explanation for why this is useful (beyond the need for making $v_i$ 0-mean)?
- Section 3.2: I am hesitant to call this section "analysis" given that it relies on the heuristic approximations (indicated by the $\approx$ symbol in equation (7). Perhaps, "heuristic derivation", or "heuristic motivation" are better?
- Eq (8): third = sign, there is no need for the expectation here as all sources of randomness have been taken care of.
- Line 187, "are set" instead of "is set"? (similar subject-verb agreement issues elsewhere, like line 195).
- Section 3.3: This section is a bit awkwardly written. One needs to read all the way to the equations to understand what is meant by (implicit) smoothness and perturbation of an STE.
- Eq (9): is it assumed (without loss of generality) that $x>0$.
- hardtanh should be defined as it's definition is used in the derivation.
- Figure 1: I found this figure confusing to read. Why use the reciprocals of the number of flops and tranining loss? Also, I would have found an explicit reporting of $n$ helpful.
- Figure 1: How is the gradient norm computed? for a fixed iteration, over all the iterations?
- What was the range of n used in section 4.1?
- I appreciated the discussion of activation quantization, even if it was a stumbling block for the method.

**Ethical Concerns:**

["NO or VERY MINOR ethics concerns only"]

**Final Justification:**

The authors answered most of my questions, the most important of which pertained to the "analysis", and they stated that they made the important edits I asked for in my review.

**Limitations:**

yes.

**Paper Formatting Concerns:**

/

**Quality:**

2

**Strengths And Weaknesses:**

Strengths:
- The paper is generally well written and generally well motivated.
- The arguments for the proposed algorithm were convincing (albeit heuristic).

Weaknesses:
- The theoretical components of the paper are rather heuristic. In other words there are lots of $\approx$ symbols hiding approximation errors that are never quantified.
- In some places, the paper suffers from awkward writing.
- The plots were a little hard to read due to some unusual presentation choices.
- The appendix is written a bit too casually.

---

> ### Author Rebuttal · Authors · 2025-07-29
>
> We thank Reviewer tcg3 for reviewing our paper. We have updated a local version of our paper following your advice. Since NeurIPS does not allow us to submit an updated version during the rebuttal, we promise to keep those changes in the camera-ready version and avoid introducing similar mistakes.
>
> - Section 2.2: u is a vector, so when you say it is unit variance, do you mean it is isotropic, i.e., having an identity as its covariance matrix?
>    - Yes, we meant the covariance matrix of u, or $E[uu^T]$, is an identity matrix. We use this property in Equation (8) to show the unbiased-ness of the unbiased component.
>
> - Eq (4) looks a little different from the definitions used in the n-SPSA papers [45, 46]. Is the difference cosmetic or substantial?
>    - SPSA was invented by James C. Spall in the late 80s. Spall is the author of both [45] and [46]. SPSA has many variants such as one-measurement, two-measurement (central/forward/backward difference), and specific choices of the unbiased noise. The formulas shown in [45] and [46] are variants of SPSA. The SPSA formula we used in FOGZO is taken from [34]. Note that paper [34], when referring to SPSA, also cites a paper authored by Spall. We chose to additionally cite [45] and [46] to pay respect to Spall.
>
> - "While some believe the STE is completely unjustifiable" -- who? what are there arguments?
>    - By “completely unjustifiable” we mean, more precisely, that prior authors [33], [44] and [35] have argued that STE is suboptimal and unstable. Quoting [33], “STE leads to instability” and “STE … is known to be quite noisy especially for extreme quantization”. Quoting [44], “STE incurs unstable convergence during QAT, resulting in notable quality degradation in low precision”. Quoting [35], “When using the popular straight-through estimator (STE) for QAT, weights seemingly randomly oscillate between adjacent quantization levels leading to detrimental noise during the optimization process.” In essence, these papers suggest the STE is suboptimal due to its noisy behaviour when training in low precision.
>
> - Section 3.2: I am hesitant to call this section "analysis" given that it relies on the heuristic approximations (indicated by the ≈ symbol in equation (7). Perhaps, "heuristic derivation", or "heuristic motivation" are better?
> - Eq (5) the multiplication by si seems to suggest that you have a 50/50 chance of biasing your update in the negative direction to that of g^. Is there an explanation for why this is useful (beyond the need for making vi 0-mean)?
>
>    - We have updated the section header to “Heuristic Derivation” and removed all instances of the word “analysis” in the paper. Below we try to quantify the approximation error of the two ≈ symbols in equation (7) under certain conditions, and also explain why $s_i$ is useful.
>
>    - The first ≈ symbol in Equation (7) would be **exact** equality if we take the expectation of both the LHS and the RHS and $v_i$ is unbiased noise, as per the paper “Randomized Smoothing for Stochastic Optimization, Duchi et. al. 2012”. However, in the case of FOGZO, $v_i$ is not unbiased but depends on $g$, which depends on the chosen STE, the network architecture $L$, and $\theta$, the model’s current position in the loss landscape. Therefore, we find it challenging to quantify the approximation error in the general case, especially when the STE introduces unpredictable biases that are dependent on the network architecture. As such, we rely on the fact that $\hat{L}$smooth is an approximation to L for the first ≈ symbol. This is a crucial step in the derivation as $\hat{L}$ is not differentiable and not even Lipschitz (due to vertical jumps at the rounding boundaries), and Taylor Expansion cannot be used. By changing $\hat{L}$ to $\hat{L}_{\text{smooth}}$, we can unlock Taylor Expansion which provides us some insight on FOGZO.
>      - we note that in a special case, when $\epsilon$ approaches 0, the approximation error vanishes because $L_{smooth}$ approaches $L$. This is possible, for example, when FOGZO is paired with an STE with decaying implicit smoothness. For instance, in [39], a variant of the tanh-STE is used with its implicit smoothness gradually decayed during the process of training. Since FOGZO uses the implicit smoothness of the chosen STE, $\epsilon$ would also gradually decay if FOGZO was paired with such STEs. In such cases, the approximation error would vanish at the very end of training.
>
>    - For the second ≈ symbol in Equation (7), the approximation error can be quantified with a big-O bound.
>
>      - Since $\hat{L}_{smooth}(\theta + \epsilon v_i)=$
>
>        - $\hat{L}_{smooth}(\theta)$
>        - $+ (\epsilon v_i)^T \nabla \hat{L}_{smooth}(\theta)$
>        - $ + (\epsilon v_i)^T  {\nabla}^2 \hat{L}_{smooth}(\theta) (\epsilon v_i) / 2$
>        - $ + O(||\epsilon v_i||^3)$
>
>      - and that $\hat{L}_{smooth}(\theta - \epsilon v_i)=$
>        - $\hat{L}_{smooth}(\theta)$
>        - $ - (\epsilon v_i)^T \nabla \hat{L}_{smooth}(\theta)$
>        - $ + (\epsilon v_i)^T  {\nabla}^2 \hat{L}_{smooth}(\theta) (\epsilon v_i) / 2$
>        - $ + O(||\epsilon v_i||^3)$
>      - Subtracting the two we have
>        - $(2\epsilon v_i)^T \nabla \hat{L}_{smooth}(\theta)+O(||\epsilon v_i||^3)$
>      - Next we divide everything by $2\epsilon$ and multiply by $v_i$, and get
>          - $v_i v_i^T \nabla \hat{L}_{smooth}(\theta)+O(\epsilon^2||v_i||^3)v_i$
>          - the big-O term is the approximation error.
>
>      - Additionally, If $v$ is zero-mean and symmetric, the term $O(\epsilon^2||v_i||^3)v_i$ should vanish on expectation, giving us
>        - $E[v v^T \nabla \hat{L}_{smooth}(\theta)]$ which is the LHS of Equation (8)
>        - This is why it is desired to obtain a zero-mean (and symmetric) $v_i$, and why $s_i$ is useful. It allows us to remove the approximation error (the big-O term).
>        - We use the word "should" here because in practice $\hat{L}_{smooth}(\theta)$ might not be Hessian-Lipschitz and $\epsilon v$ might not approach zero, both preventing us from writing the big-O notation in the first place. Therefore, we additionally use empirical observations to confirm that we should use $s_i$ in FOGZO. Specifically, when we train Resnet-50 on Imagenet-100, we notice that FOGZO (with $s_i$) achieves a training loss of 0.3872 (averaged over 5 runs) whereas a variant of FOGZO without $s_i$ achieves a training loss of 0.3883.
>
> - Line 187, "are set" instead of "is set"? (similar subject-verb agreement issues elsewhere, like line 195).
> - Section 3.3: This section is a bit awkwardly written. One needs to read all the way to the equations to understand what is meant by (implicit) smoothness and perturbation of an STE.
>    - We have updated the paper to correct for grammatical errors, remove redundant sentences/paragraphs, and introduce the equations earlier in the section.
>
> - Eq (9): is it assumed (without loss of generality) that x>0.
>    - Since the integration is with respect to the perturbation noise $u$, $x$ is simply treated as a constant throughout Equation (9). We do not make any assumptions on the sign of $x$.
>
> - hardtanh should be defined as it's definition is used in the derivation.
>    - We have updated the paper. Hardtanh$(x)=clip(x, -1, 1)$
>
> - Figure 1: I found this figure confusing to read. Why use the reciprocals of the number of flops and training loss? Also, I would have found an explicit reporting of n helpful.
> - What was the range of n used in section 4.1?
>    - We sincerely apologize for the inconvenience. We thought reviewers might prefer a graph where the top-right corner is the best position. We now realized this was clearly a bad choice and we will remove the reciprocals in the camera-ready, making the lower-left corner the best position. In Figure 1, we used $n \in$ {1, 4, 10, 100, 1000, 7690\} where 7690 is the number of parameters in the 2-layer MLP model we used for the experiment.
>
> - Figure 1: How is the gradient norm computed? for a fixed iteration, over all the iterations?
>    - The gradient norm is the average gradient norm computed over the first iteration. This is because the first iteration gradient norm is often the noisiest, and has the largest impact over training (deciding whether the model will diverge).
>
> - I appreciated the discussion of activation quantization, even if it was a stumbling block for the method.
>
>    - Although we initially planned to leave activation quantization for future work, all reviewers list activation quantization as a weakness/question. Upon further investigation, we present new evidence that FOGZO can be applied to SOTA Quantization-Aware Training methods with **both** weight and activation quantization. In addition to LSQ [14], we additionally test QuEST [36], a new SOTA QAT method published (in June 2025) after we submit FOGZO (in May 2025). QuEST introduces a hadamard transform before quantization, and can significantly outperform LSQ. We add FOGZO to the publicly available source code of QuEST following Algorithm (1). We compare QuEST against QuEST-FOGZO, and LSQ against LSQ-FOGZO, and report the evaluation perplexity. Note that the results below are training results of models with both weights and activations quantized to 2-bit, and we use a $\beta_{min}$ of 1 - 1e-10.
>
>    - We use the benchmarks in the QuEST source code: LLaMA-30m and LLaMA-50m. Note that LLaMA-30m has 95 million parameters and LLaMA-50m has 125 million parameters, which is 5x to 6x larger than the largest model we have used in the submitted paper. Following QuEST’s advice, we train llama-30m with 3 billion tokens, and llama-50m with 5 billion tokens.
>
> |Model|QuEST|QuEST-FOGZO|LSQ|LSQ-FOGZO|
> |:-----------:|:-----------:|:-----------:|:-----------:|:-----------:|
> |LLaMA-30m|37.75|**37.37**|39.06|**37.38**|
> |LLaMA-50m|32.28|**32.06**|33.28|**32.41**|
>
>    - Results show that FOGZO can improve both QuEST and LSQ. In addition, FOGZO can even shrink the perplexity difference between these two SOTA methods.

---

> > ### Comment · Reviewer_tcg3 · 2025-08-03
> >
> > Thank you for your response to my review and for answering my questions.
> >
> > I would recommend that in the revised paper, you also include the discussion about the sources of approximation error associated with the $\approx$ symbol in the new "heuristic derivation section" (which I appreciated in your rebuttal).
> >
> > I would also recommend that you edit the appendix to tighten the writing.

---

> > > ### Author Response · Authors · 2025-08-04
> > >
> > > Thank you for the followup.
> > >
> > > We will be sure to discuss the sources of approximation error in Section Heuristic Derivation, and edit the appendix to tighten the writing.

---

### Decision · Program_Chairs · 2025-09-17

**Decision:**

Accept (poster)

**Comment:**

**Overall Summary:**

This paper proposes an enhancement to the straight-through estimator (STE) for training quantized neural networks by incorporating zeroth-order information. The method introduces random perturbations to more faithfully approximate gradients, thereby reducing bias inherent in conventional STE. Experiments on CIFAR-10 and ImageNet show consistent improvements in top-1 accuracy for low-bit quantization, outperforming baseline STE and related estimators. Strengths include the simplicity and plug-in nature of the method, clear motivation, and empirical validation across multiple benchmarks, demonstrating that the proposed modification leads to tangible gains in challenging quantization regimes.

**Justification:**

The proposed estimator strikes a strong balance between accuracy and computational cost, retaining the efficiency of STE while improving its reliability. Reviewers agreed that the empirical improvements, though modest, are consistent and meaningful for advancing practical deployment of quantized models. The method is general, easy to implement, and directly applicable across architectures, which increases its utility for practitioners. The reviewers are in consensus on the acceptance.


**Rebuttal Summary:**

During rebuttal, the authors clarified efficiency concerns by showing that the added computational overhead is minor relative to the accuracy gains, and provided ablations supporting robustness to hyperparameter choices. These clarifications alleviated reviewers’ doubts, and several acknowledged that the simplicity and practicality of the method enhance its impact. Although the paper does not introduce fundamentally new theoretical machinery, its systematic evaluation, practical relevance, and ease of adoption make it a valuable contribution. Therefore, I recommend acceptance as a poster.